# Linear Model Merging Unlocks Simple and Scalable Multimodal Data Mixture Optimization

## Abstract

Selecting the best data mixture is critical for successful Supervised Fine-Tuning (SFT) of Multimodal Large Language Models. However, determining the optimal mixture weights across multiple domain-specific datasets remains a significant bottleneck due to the combinatorial search space and the high cost associated with even a single training run. This is the so-called Data Mixture Optimization (DMO) problem. On the other hand, model merging unifies domain-specific experts through parameter interpolation. This strategy is efficient, as it requires only a single training run per domain, yet it often leads to suboptimal models. In this work, we take the best of both worlds, studying model merging as an efficient strategy for estimating the performance of different data mixtures. We train domain-specific multimodal experts and evaluate their weighted parameter-space combinations to estimate the efficacy of corresponding data mixtures. We conduct extensive experiments on 14 multimodal benchmarks, and empirically demonstrate that the merged proxy models consistently select near-optimal mixtures. This decouples the search for optimal mixtures from the resource-intensive training process, thereby providing a scalable and efficient strategy for navigating the complex landscape of mixture weights. Code and models will be publicly available.

## 1 Introduction

Supervised fine-tuning (SFT) (Liu et al., 2023; Dai et al., 2023) is a post-training strategy to instill instruction-following abilities into Multimodal Large Language Models (MLLMs) and improve their performance on downstream tasks. SFT finetunes the pretrained model on a diverse mixture of high-quality instruction-tuning datasets covering various visual tasks, such as visual question answering, optical character recognition, and counting (Tong et al., 2024; Chen et al., 2025a). The proportions of data allocated to each task, i.e., the *mixture weights*, have a critical impact on the resulting model (Albalak et al., 2023; Liu et al., 2025; Xie et al., 2023). As a consequence, most state-of-the-art MLLMs put strong emphasis on optimizing the data mixture, a problem called Data Mixture Optimization (DMO). This problem is challenging as evaluating a set of mixture weights requires the computationally expensive fine-tuning on the corresponding data mixture. Thus, most models rely on balanced sampling or trial-and-error approaches (Li et al., 2025a; Tong et al., 2024; Deitke et al., 2025), and even advanced strategies, such as fitting scaling laws (Li et al., 2025c; Shukor et al., 2025) or supervised regression (Liu et al., 2025), cannot avoid several, costly fine-tuning runs.

As opposed to SFT on a mixture of multiple domains, model merging represents another viable strategy to achieve the same goal of consolidating knowledge from various sources (Wortsman et al., 2022a; Ilharco et al., 2023). Merging acts at the parameter level, combining the weights of multiple *expert* models fine-tuned on separate domain-specific data. Thus, it requires as many fine-tuning runs as there are domains, being naturally much less expensive than even the cheapest approaches for DMO. However, it inevitably depends on a variety of design choices concerning *how* to merge models, which often lead to merged models that underperform those fully fine-tuned on unified, multi-domain mixtures (Yadav et al., 2023; Jin et al., 2023).

In this work, we seek to bridge these two paradigms by investigating whether the efficiency of model merging can be leveraged for Data Mixture Optimization (DMO). Our central hypothesis is that, for any set of

mixture weights, a linear combination of domain experts can serve as an effective surrogate for a model fully fine-tuned on data mixed with the same weights. To test this hypothesis, we first train expert models for individual domains, then merge them according to candidate domain ratios, and treat the resulting merged models as proxies for the corresponding mixture-trained models. We use the performance of these proxies to estimate downstream accuracy, rank candidate mixtures, and ultimately select the optimal one. This approach enables exploring the grid of combinations only at evaluation, as opposed to exploring the grid at training time. Previous work explored the idea of parameter merging as a proxy for data mixing in unimodal tasks (Maldonado et al., 2024; Tao et al., 2025; Wang et al., 2026). Instead, we focus on multimodal LLM supervised fine-tuning, where data sources correspond to heterogeneous vision-language domains.

We conduct an extensive empirical evaluation to compare merged proxies and mixture-trained models along numerous axes of exploration: model family (Qwen2-VL (Wang et al., 2024) and Intern3.5-VL (Wang et al., 2025)), model size (2B and 8B parameters variants), fine-tuning strategy (LoRA (Hu et al., 2022) and full fine-tuning), number of domains to mix (2, 3, 4 domains), and data budget (10k, 50k, 100k samples). We additionally assess that merged proxies can serve for both *specialist* scenarios (*i.e.*, optimizing for a specific downstream task of interest) and for *generalist* scenarios, where the aim is to pick the mixture leading to the best average performance on a broad spectrum of tasks.

In general, merged proxies correlate well with mixture-trained models, and consequently enable cheap selection of near-optimal mixtures. Furthermore, we empirically observe that the DMO problem is more relevant on specialist objectives, where selecting the optimal mixture can lead to significant gains in performance. Instead, we find that differences among mixtures are moderate on the generalist objective, since gains on some tasks are offset by losses on others.

Additionally, we provide a theoretical intuition for why linearly merged models are well-behaved surrogates. This intuition builds on a second-order Taylor approximation of the loss on a given mixture under local convexity assumptions, which we empirically validate. Overall, these findings establish model merging as an effective surrogate for data mixture evaluation, opening future avenues for efficient DMO in MLLMs.

Overall, our contributions are as follows:

- We investigate model merging as a practical surrogate for optimizing data mixtures in MLLM supervised fine-tuning. The merged proxy enables evaluating the performance of diverse data mixtures with a small fixed number of training runs, replacing expensive fine-tuning.

- We perform extensive experiments and empirically verify a strong correlation between the performance of merged models and models trained directly on corresponding data mixtures, providing large-scale evidence that simple linear merging is an effective surrogate for multimodal data-mixture selection.

- We provide a theoretical justification of the efficacy of the proposed merged proxy, grounded in a second-order approximation of the fine-tuning loss.

- We publicly release over 150 trained model checkpoints, providing a resource for future research on data mixture optimization.

## 2 Related Work

**Data Mixture Optimization (DMO)** refers to the problem of selecting optimal mixture weights to sample different data sources during training. Despite its straightforward formulation, DMO represents a challenging and expensive problem in practice, as naive solutions would require end-to-end training to evaluate each candidate mixture, as done in Tong et al. (2024). Older approaches focus on optimizing the worst-case domain loss (Xie et al., 2023), while some of the most prominent modern strategies try to alleviate the cost of DMO by fitting supervised models to regress mixture weights into reference performance targets after sampling a population of training runs. Such supervised models can take the form of power laws (Shukor et al., 2025; Ye et al., 2025), as well as simple linear or tree regression (Liu et al., 2025). However, while surely less expensive than naive grid search, these approaches still require tens or even hundreds of training runs (Wettig et al., 2025). Recent work also study mixture optimization specifically for multimodal training (Xie et al., 2026).

In this work, we demonstrate that model merging offers a simple yet effective strategy for efficiently tackling DMO in MLLM supervised fine-tuning , despite requiring as many training runs as there are domains.

**Model Merging** aims to unify the parameters of multiple fine-tuned models with a shared architecture into a single model (Wortsman et al., 2022a). This principle has been applied to many downstream tasks where models are trained on different sets of data, such as federated learning (McMahan et al., 2017; Singh & Jaggi, 2020; Wang et al., 2020), OOD generalization (Izmailov et al., 2022; Gupta et al., 2020; Cha et al., 2021), continual learning (Wortsman et al., 2022b; Liu & Soatto, 2023; Dziadzio et al., 2025), and, more recently, to combine the capabilities of multiple multimodal language models (Chen et al., 2025b; Qu et al., 2025; Du et al., 2025; Sung et al., 2023). By definition, merging techniques vary depending on *how* different parameter sets are combined into one. For instance, Matena & Raffel (2022) proposes a Fisher-informed linear combination. Yadav et al. (2023) proposes a three-stage approach, modeling interference between parameters of different models. Yu et al. (2024) performs model merging via sparsification and parameter scaling. Gargiulo et al. (2025) uses singular value decomposition of task matrices (Ilharco et al., 2023) to model task interference. In contrast, we do not introduce a new merging technique. Instead, we show that model merging is an effective strategy for data mixture optimization, avoiding costly per-mixture training.

**Model Merging and Data Mixing.** Several recent works analyze the relation between parameter merging and data-level mixing. One line of research compares data mixing and model merging as alternative ways to combine capabilities for LLM alignment across helpfulness, honesty, and harmlessness (Yang et al., 2026), multilingual multi-task learning (Ahmadian et al., 2024), and multi-task code LLMs (Zhu et al., 2026). Closest to our work, another line of research studies parameter merging as a low-cost proxy for selecting task or mixture weights. Notably, Maldonado et al. (2024) proposes using merged task-specific models as fast previews for selecting weights in multitask fine-tuning, and provide a Bayesian interpretation connecting task-loss weights and merge coefficients. Tao et al. (2025) shows that merging is indicative of whether a data source shall be added or removed from the training set of a model. This problem setting differs from ours, which instead operates at a fixed training budget, and is not realistic in MLLM applications as it confounds data mixing with the benefits of adding more data. More recently, Wang et al. (2026) explores model merging as a proxy for DMO in text-only LLM fine-tuning. In this work, we extend this line of research to multimodal data mixture optimization for MLLM supervised fine-tuning, showing that simple linear merging can serve as a practical surrogate for selecting training mixtures across MLLM families, scales, domain counts, fine-tuning regimes, and both specialist and generalist objectives.

## 3 Data Mixture Optimization via Model Merging

In this section, we formalize the problem of Data Mixture Optimization for Multimodal LLMs, and we describe how model merging can be used as a proxy to estimate the effect of different mixture weights.

### 3.1 Data Mixture Optimization

**Setting.** We consider MLLMs with the standard architecture ViT $\rightarrow$ Adapter $\rightarrow$ LLM, where an adapter module aligns the visual features from a pretrained encoder (Dosovitskiy et al., 2021) to the input space of a Large Language Model. The training pipeline of a general-purpose MLLM typically consists of (i) a pre-training stage, where the adapter's weights are optimized to align the two modalities, followed by (ii) a supervised fine-tuning (SFT) stage on instruction data spanning diverse domains.

Let $\{\mathcal{D}_1, \ldots, \mathcal{D}_K\}$ denote $K$ domain-specific SFT datasets and let $N$ be a fixed training budget (*i.e.*, number of data points). A data mixture

$$\mathcal{D}_\mathbf{w}(N) = \bigcup_{i=1}^{K} w_i \mathcal{D}_i, \tag{1}$$

with mixing weights $\mathbf{w} = (w_1, \ldots, w_K)$ in the probability simplex $\Delta^{K-1}$, is a collection of $N$ unique samples, where each sample is drawn from domain $\mathcal{D}_i$, after sampling the domain index $i$ with probability $w_i$. Since large collections of SFT data are publicly available (Tong et al., 2024; Wiedmann et al., 2025; Guo et al., 2025), we assume that each original dataset has $|\mathcal{D}_i| \geq N$ diverse samples. Consequently, the mixture $\mathcal{D}_\mathbf{w}(N)$

contains approximately $w_i N$ distinct samples from domain $\mathcal{D}_i$. Since the data budget $N$ is fixed, we will omit it from the notation in the following.

Supervised fine-tuning a base model $\boldsymbol{\theta}_0$ on mixing ratios $\mathbf{w}$ leads to the model

$$\boldsymbol{\theta}_\mathbf{w}^* = \operatorname*{argmin}_{\boldsymbol{\theta}} \mathcal{L}(\boldsymbol{\theta}, \mathcal{D}_\mathbf{w}), \tag{2}$$

where $\mathcal{L}(\boldsymbol{\theta}, \mathcal{D}_\mathbf{w})$ is the empirical training loss.

**Optimization objective.** Let $f : \boldsymbol{\theta} \mapsto \mathbb{R}$ be a performance measure. We are interested in the dependence of performance on the data mixture:

$$f(\mathbf{w}) : \mathbf{w} \mapsto f(\boldsymbol{\theta}_\mathbf{w}^*) \in \mathbb{R}. \tag{3}$$

In particular, Data Mixture Optimization (DMO) consists of finding the optimal mixing weights:

$$\max_\mathbf{w} f(\mathbf{w}). \tag{4}$$

In practice, there are diverse meaningful choices for the performance measure. In this work, we will focus on both task-specific and general-purpose objectives. For the former, we measure performance against a benchmark of interest (*i.e.*, the "task"). For the latter, we measure the average performance on a wide collection of downstream tasks.

**Computational challenge.** Evaluating the performance $f(\mathbf{w})$ of a certain mixing ratio requires fine-tuning the base model on the corresponding data mixture. This is computationally expensive, making the DMO problem challenging in practice. Even a coarse grid over mixture ratios leads to a number of training runs that grows exponentially with the number of domains. Our goal is therefore to estimate relative performance across mixtures without retraining a model for each candidate $\mathbf{w}$.

## 3.2 Model Merging for Efficient DMO

Model merging combines the parameters of multiple fine-tuned models to obtain a single model with aggregated capabilities. Existing work typically studies merging as a way to directly obtain a high-performing multitask model (Yadav et al., 2023; Yu et al., 2024; Matena & Raffel, 2022; Gargiulo et al., 2025). Here, we instead use merging as a performance proxy for DMO.

We first obtain $K$ *experts* models $\boldsymbol{\theta}_i = \boldsymbol{\theta}^*(\mathcal{D}_i)$, $i = 1, \ldots, K$, by fine-tuning the base model $\boldsymbol{\theta}_0$ on a single domain $\mathcal{D}_i$. Then, for a candidate mixture $\mathbf{w}$, we approximate the mixture-trained model $\boldsymbol{\theta}_\mathbf{w}^*$ with a merged model

$$\boldsymbol{\theta}_\mathbf{w}^M = w_1 \boldsymbol{\theta}_1 + \cdots + w_K \boldsymbol{\theta}_K. \tag{5}$$

We then use $f(\mathbf{w}) \approx f(\boldsymbol{\theta}_\mathbf{w}^M)$ as a surrogate for the true performance under mixture $\mathbf{w}$. This reduces the cost of evaluating any mixture from a *full* training run to a *single* evaluation, after the $K$ expert models have been trained.

---

**Algorithm 1** DMO via Model Merging

```
# Functions:
# train(model, D) = returns model trained on dataset D
# score(model, target) = returns scalar performance of model on
      target
# merge(models, weights) = returns linearly merged model

def dmo_via_merging(base_model, datasets, mixture_candidates, target)
    :
    # Arguments:
    # base_model = Base Model for Supervised Fine-Tuning
    # datasets = list of [D_1, ..., D_K]
    # mixture_candidates = list of candidate mixtures
    # target = performance target to maximize

    # step 1: train experts
    experts = [train(base_model, dataset) for dataset in datasets]

    # step 2: evaluate merged proxies
    scores = [
        score(merge(experts, mixture_weights), target)
        for mixture_weights in mixture_candidates
    ]

    # step 3: select mixture according to merged proxies
    return mixture_candidates[scores.index(max(scores))]
```

---

Importantly, the merged model $\boldsymbol{\theta}_\mathbf{w}^M$ is not expected to match $\boldsymbol{\theta}_\mathbf{w}^*$ in parameter space or absolute performance. For DMO, we only require that the surrogate preserves the ordering of mixtures, *i.e.*, that performance under merging is monotonically related to the true performance:

$$f(\boldsymbol{\theta}_{\mathbf{w}_1}^M) \leq f(\boldsymbol{\theta}_{\mathbf{w}_2}^M) \text{ whenever } f(\boldsymbol{\theta}_{\mathbf{w}_1}^*) \leq f(\boldsymbol{\theta}_{\mathbf{w}_2}^*). \tag{6}$$

Table 1: SFT data collection comprising 23 datasets belonging to 4 categories: General Multimodal Understanding, Optical Character Recognition, Visual Perception & Counting, and Charts Understanding.

**General Multimodal Understanding (General)**
ALLaVA-Instruct (LAION) (Chen et al., 2024a), VQAv2 (Goyal et al., 2017), lnQA (Pont-Tuset et al., 2020), LVIS-Instruct-4v (Wang et al., 2023), Q-Align (Wu et al., 2024), GQA (Hudson & Manning, 2019), VizWiz (Gurari et al., 2018), Visual7W (Zhu et al., 2016)

**Optical Character Recognition (OCR)**
SynthDog Modified (Kim et al., 2022), OCR-VQA (Mishra et al., 2019), DocVQA (Mathew et al., 2021), TextVQA (Singh et al., 2019), TextCaps (Sidorov et al., 2020), LLaVAr (Zhang et al., 2023), ST-VQA (Biten et al., 2019), Rendered-Text (Wendler, 2023), InfoVQA (Mathew et al., 2022)

**Visual Perception & Counting (Vis. Perc.)**
Clevr (Johnson et al., 2017), TallyQA (Acharya et al., 2019)

**Charts Understanding (Charts)**
dVQA (Kafle et al., 2018), ChartQA (Masry et al., 2022), Chart2Text (Kantharaj et al., 2022), VisText (Tang et al., 2023)

Thus, merging is used as a ranking proxy rather than as a method for constructing the final model. For this reason, we refer to merged models as *merged proxies*. Given a set of candidate mixture weights $\mathcal{W} \subset \Delta^{K-1}$ we then estimate the optimal mixture as $\arg\max_{\mathbf{w} \in \mathcal{W}} f(\boldsymbol{\theta}_{\mathbf{w}}^M)$. Algorithm 1 provides a summary of this simple procedure.

Note that, in principle, any merging technique could be applied within this framework. While several methods have been proposed to maximize the performance of merged models, our objective is different: we seek a stable and informative proxy for mixture evaluation. In section 4, we empirically show that simple linear weight averaging is sufficient to obtain strong rank correlation with true mixture performance. Additionally, in section 5, we provide a theoretical analysis that supports the experimental observation.

## 4 Experimental Evaluation

In this section, we conduct experiments to quantify the effectiveness of using merged proxies to rank data mixtures. We first highlight which models we fine-tune, the data collection for SFT, and the selected suite of benchmarks. Then, we show that merged proxies successfully select near-optimal mixtures regardless of (i) number of domains (section 4.1) and (ii) model size (section 4.2). Finally, we provide a comparison to regression-based approaches (Liu et al., 2025), the *de facto* standard for DMO, highlighting their inefficiency *w.r.t.* merged proxies (section 4.3).

**Base Models.** We conduct all experiments starting from base models from both the Qwen2-VL (Wang et al., 2024) and Intern3.5-VL (Wang et al., 2025) families. We deliberately choose to use a recent model (Intern3.5-VL) alongside a less recent one (Qwen2-VL), as this allows us to study both (i) models from different families and (ii) different pre-training recipes. This is because the pre-training mixture of Intern3.5-VL largely contains instruction tuning data, while the pre-training mixture of Qwen2-VL does not.

**Data for Supervised Fine-Tuning.** We collect a corpus of 23 datasets for SFT, categorized under 4 different domain tags: General Multimodal Understanding, Optical Character Recognition (OCR), Visual Perception & Counting, and Charts Understanding. Table 1 lists all the different data sources, grouped by category. For each category, we construct a domain-specific dataset of 100k samples by uniformly sampling the corresponding data sources. For all datasets with multiple splits, we employ the *train* split to avoid contamination with target benchmarks.

**Target Benchmarks.** We select 14 benchmarks to measure agreement between merged proxies and mixture-trained models. These benchmarks cover the four categories represented in the training mixtures, and overall provide a measurement of the general performance of a model. We validate against GQA (Hudson & Manning, 2019), OK-VQA (Marino et al., 2019), VQAv2 (Goyal et al., 2017), VizWiz (Gurari et al., 2018), DocVQA

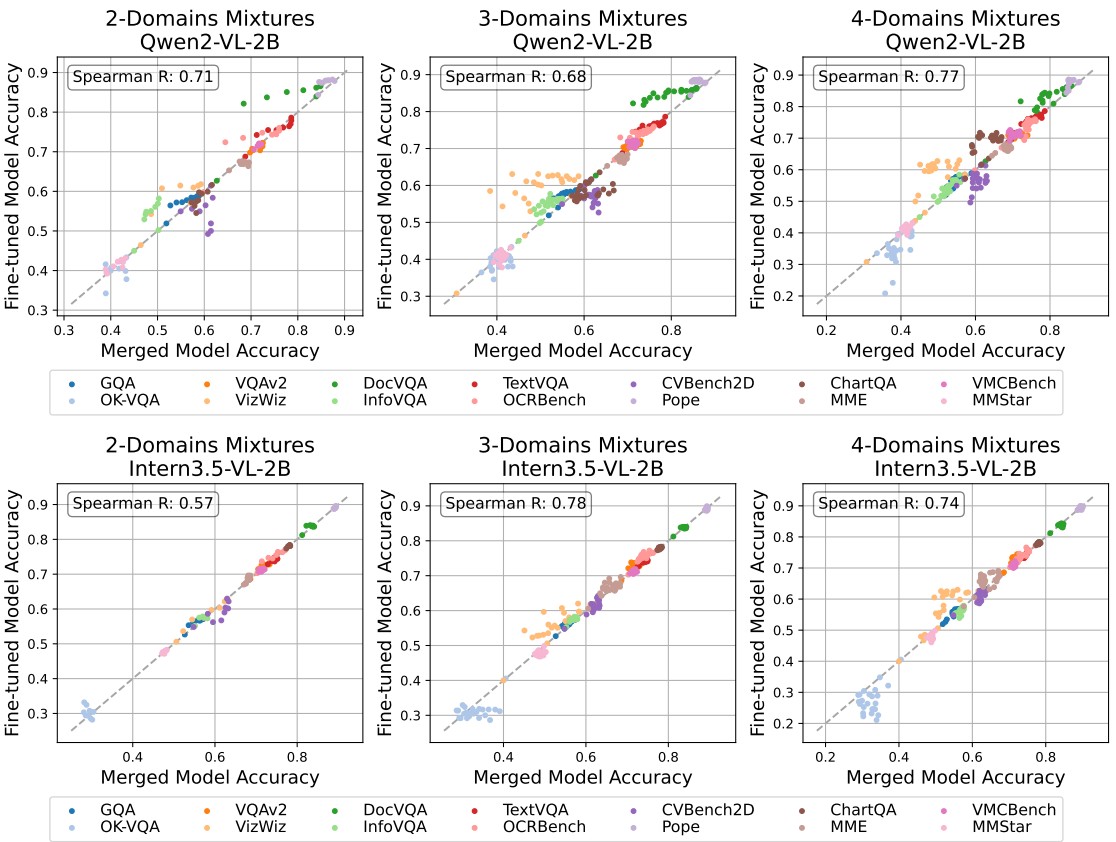

Figure 1: Correlation plots between downstream accuracies of mixture-trained models and our proposed merged proxy. Results are shown for Qwen2-VL-2B and Intern3.5-VL-2B models, fine-tuned on $2, 3, 4$-domains data mixtures. Each plot reports the Spearman's rank correlation coefficient (R) of the average performance.

(Mathew et al., 2021), InfoVQA (Mathew et al., 2022), TextVQA (Singh et al., 2019), OCRBench (Liu et al., 2024), CVBench2D (*Counting* subtask) (Tong et al., 2024), Pope (Li et al., 2023), ChartQA (Masry et al., 2022), MME (Fu et al., 2025), VMC-Bench (Zhang et al., 2025b), and MMStar (Chen et al., 2024b).

**Implementation Details.** Models are trained with the AdamW optimizer (Loshchilov & Hutter, 2019) with a peak learning rate of $2 \times 10^{-5}$, linearly warmed up during the initial 10% of steps and decayed with a cosine schedule. We fix the batch size to 128. Unless otherwise specified, we train medium-sized 2B models with a fixed data budget of 100k samples, and apply low-rank adaptation (Hu et al., 2022) to all linear projections of the LLM with a rank of 16. In appendix C.3 we report additional results where we fully fine-tune both models. Training runs are performed with `llama-factory` (Zheng et al., 2024) and evaluations with `lmms-eval` (Zhang et al., 2025a).

## 4.1 Merged models are effective proxies

We begin our experiments by evaluating the performance of merged proxies as the optimization landscape becomes more complex, *i.e.*, by gradually increasing the number of domains from 2 to 4. For each setting, we consider the following domains for the data mixture:

1. **For $K=2$ domains**, we consider General + OCR datasets, as they are naturally more abundant. In this setup, we consider the grid of all valid mixtures with a step size of 1/8, which results in a total of 7 mixtures.
2. **For $K=3$ domains**, we add Visual Perception. As for 2-domain mixtures, we consider a grid with a step size of 1/8, which contains 21 valid mixtures.

Table 2: Downstream accuracies of mixture-trained models. We compare the performance of the mixture selected by the merged proxy (**Selected**) against the **Uniform** mixture, **Median** performance among candidate mixtures, and the **Best** mixtures (the upper bound obtained via grid search). The **Average Performance** row corresponds to *generalist* scenarios, where the goal is to select a mixture that performs well overall. Other rows correspond to *specialist* cases, where the mixture is optimized for a downstream task. For **Selected**, we report the average accuracy and the standard deviation (subscript) over three runs.

| Target Benchmarks | $K = 2$ DOMAINS | | | | $K = 3$ DOMAINS | | | | $K = 4$ DOMAINS | | | |
|---|---|---|---|---|---|---|---|---|---|---|---|---|
| | Uniform | Median | Selected | Best | Uniform | Median | Selected | Best | Uniform | Median | Selected | Best |
| *Qwen2-VL-2B* | | | | | | | | | | | | |
| GQA | 57.8 | 57.8 | $58.8_{0.0}$ | 58.8 | 57.8 | 57.5 | $58.7_{0.0}$ | 58.7 | 56.9 | 57.4 | $58.4_{0.0}$ | 58.4 |
| OK-VQA | 40.7 | 40.1 | $38.2_{0.8}$ | 41.6 | 42.5 | 40.3 | $38.4_{0.6}$ | 42.2 | 35.8 | 34.5 | $38.5_{2.0}$ | 40.7 |
| VQAv2 | 71.4 | 71.4 | $72.1_{0.3}$ | 72.2 | 71.8 | 71.0 | $71.6_{0.4}$ | 72.1 | 70.5 | 70.4 | $71.0_{0.2}$ | 71.6 |
| VizWiz | 61.0 | 61.0 | $60.4_{1.1}$ | 61.8 | 62.9 | 61.1 | $62.5_{0.0}$ | 63.1 | 62.0 | 61.0 | $58.5_{0.0}$ | 63.0 |
| DocVQA | 85.5 | 85.5 | $86.9_{0.0}$ | 86.9 | 85.5 | 85.4 | $86.3_{0.1}$ | 86.4 | 85.1 | 84.1 | $86.6_{0.0}$ | 86.6 |
| InfoVQA | 55.0 | 55.0 | $58.2_{0.0}$ | 58.2 | 55.1 | 55.7 | $56.3_{0.2}$ | 57.7 | 54.1 | 54.6 | $58.0_{1.0}$ | 58.4 |
| TextVQA | 76.1 | 76.1 | $77.7_{0.4}$ | 77.9 | 76.4 | 76.2 | $77.0_{0.1}$ | 77.2 | 76.6 | 75.8 | $76.6_{0.4}$ | 77.5 |
| OCRBench | 74.3 | 74.4 | $75.3_{0.0}$ | 75.3 | 74.8 | 74.4 | $75.5_{0.0}$ | 75.5 | 75.2 | 74.8 | $75.5_{0.1}$ | 76.0 |
| CVBench2D-Count | 49.2 | 54.9 | $54.1_{3.6}$ | 58.4 | 58.0 | 57.2 | $56.8_{2.1}$ | 59.9 | 57.9 | 56.8 | $56.9_{0.6}$ | 61.3 |
| Pope | 88.0 | 88.0 | $88.2_{0.0}$ | 88.2 | 88.3 | 88.4 | $88.4_{0.1}$ | 88.7 | 88.3 | 88.2 | $88.5_{0.0}$ | 88.6 |
| ChartQA | 57.8 | 57.8 | $59.9_{0.0}$ | 59.9 | 57.7 | 57.8 | $59.3_{0.9}$ | 60.9 | 70.3 | 70.1 | $71.6_{0.0}$ | 71.7 |
| MME | 67.3 | 67.3 | $67.3_{0.0}$ | 67.6 | 66.8 | 67.3 | $67.5_{0.5}$ | 68.4 | 67.7 | 66.9 | $67.0_{0.3}$ | 67.8 |
| VMCBench | 71.8 | 71.8 | $71.8_{0.1}$ | 72.1 | 71.1 | 71.6 | $71.6_{0.3}$ | 72.2 | 70.9 | 71.5 | $71.9_{0.5}$ | 72.6 |
| MMStar | 42.2 | 41.0 | $41.6_{0.7}$ | 42.7 | 41.2 | 40.8 | $40.7_{0.6}$ | 42.6 | 41.4 | 41.2 | $41.9_{0.3}$ | 42.7 |
| **Average Performance** | 64.1 | 64.4 | $64.6_{0.1}$ | 64.8 | 65.0 | 64.4 | $64.6_{0.1}$ | 65.1 | 65.2 | 64.8 | $65.3_{0.1}$ | 65.5 |
| *Intern3.5-VL-2B* | | | | | | | | | | | | |
| GQA | 56.8 | 56.6 | $57.2_{0.0}$ | 57.2 | 56.5 | 56.3 | $57.0_{0.0}$ | 57.0 | 56.2 | 56.4 | $56.9_{0.0}$ | 56.9 |
| OK-VQA | 32.5 | 30.3 | $28.2_{0.0}$ | 33.2 | 30.6 | 31.0 | $31.2_{0.0}$ | 32.9 | 25.2 | 26.3 | $32.2_{0.0}$ | 32.2 |
| VQAv2 | 72.6 | 72.7 | $74.1_{0.0}$ | 74.1 | 73.0 | 73.0 | $74.1_{0.0}$ | 74.3 | 73.6 | 73.2 | $73.9_{0.0}$ | 74.5 |
| VizWiz | 59.7 | 59.7 | $62.8_{0.0}$ | 62.8 | 57.9 | 57.0 | $62.0_{0.0}$ | 62.0 | 58.9 | 61.0 | $62.6_{0.0}$ | 63.0 |
| DocVQA | 84.0 | 83.9 | $83.7_{0.1}$ | 84.1 | 84.0 | 83.9 | $83.8_{0.2}$ | 84.0 | 84.3 | 83.8 | $84.0_{0.2}$ | 84.4 |
| InfoVQA | 57.4 | 57.5 | $57.5_{0.1}$ | 57.8 | 57.7 | 57.5 | $57.7_{0.3}$ | 58.3 | 56.3 | 56.3 | $57.0_{0.4}$ | 57.9 |
| TextVQA | 74.5 | 74.5 | $74.5_{0.1}$ | 74.8 | 74.4 | 74.4 | $74.4_{0.2}$ | 74.8 | 74.3 | 73.9 | $74.7_{0.0}$ | 74.7 |
| OCRBench | 75.5 | 75.5 | $76.8_{0.0}$ | 76.8 | 75.6 | 75.3 | $76.5_{0.0}$ | 77.2 | 75.0 | 74.0 | $75.3_{0.2}$ | 76.6 |
| CVBench2D-Count | 59.0 | 59.0 | $60.4_{1.0}$ | 62.2 | 61.8 | 61.2 | $64.0_{0.5}$ | 64.2 | 60.5 | 61.2 | $62.7_{0.3}$ | 64.8 |
| Pope | 89.1 | 89.1 | $89.5_{0.0}$ | 89.5 | 88.8 | 89.4 | $89.2_{0.3}$ | 89.9 | 89.3 | 89.6 | $89.7_{0.1}$ | 89.9 |
| ChartQA | 78.1 | 78.1 | $78.1_{0.0}$ | 78.4 | 77.8 | 78.0 | $78.2_{0.1}$ | 78.3 | 78.2 | 77.9 | $77.9_{0.1}$ | 78.4 |
| MME | 68.9 | 69.0 | $68.9_{0.9}$ | 69.6 | 67.1 | 66.7 | $67.4_{1.2}$ | 69.6 | 66.9 | 66.3 | $66.6_{2.3}$ | 69.1 |
| VMCBench | 71.2 | 71.3 | $71.0_{0.3}$ | 71.5 | 71.0 | 71.2 | $71.1_{0.1}$ | 71.9 | 71.3 | 71.2 | $71.4_{0.7}$ | 72.2 |
| MMStar | 48.0 | 47.6 | $48.1_{0.1}$ | 48.2 | 48.0 | 47.7 | $48.1_{0.3}$ | 48.7 | 47.7 | 47.4 | $47.3_{0.3}$ | 48.7 |
| **Average Performance** | 66.2 | 66.1 | $66.2_{0.1}$ | 66.2 | 66.0 | 65.8 | $66.3_{0.1}$ | 66.4 | 65.5 | 65.5 | $66.0_{0.0}$ | 66.0 |

3. **For $K$=4 domains**, we further add Charts to the mix. Here, we sample 20 mixtures from a multinomial Dirichlet distribution with uniform weights, thereby sampling all possible mixtures with equal probability.

We then train all models and evaluate on the aforementioned suite of 14 benchmarks. For merged proxies, we just need to fine-tune the $K$ expert models on the individual sources.

**Results** for this experiment are in figure 1, showing that the performance of merged models correlates well with that of their corresponding fine-tuned versions, with coefficients ranging from 0.57 to 0.78. Importantly, high rank correlation values hold regardless of the overall number of domains or of the model family (e.g., 0.74, 0.77 with 4 domains).

**How good are the selected mixtures?**  Rank correlation measures the agreement between mixture-trained models and merged proxies, but does not directly quantify how well the proxies perform in data mixture optimization. We therefore compare the performance of the mixture *selected* by the merged proxies with that of the *best* mixture obtained by exact grid search, which is the upper bound. As a reference, we also report results for a *uniform* mixture and the *median* performance among candidate mixtures. We consider two complementary scenarios: (i) a *specialist* setting, where the mixture is optimized for a specific downstream task, and (ii) a *generalist* setting, where the goal is to maximize average performance across all benchmarks. In the specialist case, we pick the best mixture for each of the 14 benchmarks and compare it with the mixture selected by merging proxies based on the accuracy on the same benchmark. In the *generalist* objective, we compare the mixture with the best average performance with the one selected by the merged proxies based on average performance across all benchmarks. This simulates a scenario where merging proxies are used to select a strong mixture overall.

The results for these comparisons are reported in table 2 for both model families and for $K = \{2, 3, 4\}$ domains, where **Average Performance** rows correspond to generalist scenarios and other rows to specialist ones. To evaluate the stability of the proposed method, we compute the selected mixture with 3 different random seeds and report the average accuracy across seeds, with the standard deviation in subscript. In the selection pipeline, the seed's value conditions only the experts' training, which involves randomness, while linear merging and greedy decoding during evaluation are deterministic.

**Observation #1: Merged proxies can optimize for *specialists*.** When explicitly optimizing against a target task, merged proxies perform well in either selecting the optimal data mixture or providing a candidate mixture that performs very similarly to the optimal one. In the most complex landscape with 4 domains, the selected mixture falls within $-1\%$ of exact search in 11/14 cases and 10/14 cases for Qwen2-VL and Intern3.5-VL, respectively. With 3 domains, these become 9/14 and 12/14 cases. Importantly, while some failure cases exist, successful results emerge not only for datasets whose (part of) their training sets were in the mixture, but also for different tasks such as OCR-Bench, MMStar, or MME.

**Observation #2: Merged proxies can optimize for *generalists*,** as they are capable of selecting a mixture leading to high average performance across benchmarks. Here, the mixtures proposed by the merged proxies closely match those obtained by exact grid search. Qwen2-VL with 3 domains represents the worst case, although it only displays an absolute drop of $-0.4\%$ on average performance. The best case, instead, is represented by Intern3.5-VL with 4 domains, where exactly the best mixture is chosen. However, we observe that in this setting, the choice of mixture is considerably less important than in the specialist case, as the gaps in average performance between median and best mixtures are smaller. We hypothesize that this is because the target benchmarks fall within the same domains as the training sources, implying that many different mixtures already provide sufficient task coverage. Hence, in this case, we speculate that automated data clustering techniques, which do not rely on human intuition, could be a beneficial step before optimizing mixture weights (Diao et al., 2025). More broadly, this result may also indicate that DMO becomes less critical when jointly optimizing across a large and diverse set of tasks. This observation is consistent with recent work showing that data curation methods tend to be more effective when the target distribution is narrower (Mizrahi et al., 2025; Ghosh et al., 2025).

**Observation #3: Merged proxies are scalable.** Performance remains consistently strong as $K$ increases from 2 to 4, with no significant degradation. This suggests that merged proxies are a scalable solution, despite only requiring as many training runs as there are domains.

**Observation #4: Merged proxies are stable.** The performance of the Selected mixture consistently exhibit low variability. For example, in the 4-domains setting, the average deviation in percentage points across benchmarks is 0.25 for Qwen2-VL and 0.31 for InternVL3.5-VL. On average performance it is under 0.2 in all settings. On Qwen2-VL, three benchmarks (OK-VQA, VizWiz, CVBench2D-Count) present significantly higher variability (above 1.0). Notably, these are the benchmarks where the performance of the selected mixture is significantly lower compared to the Best upper bound. This indicates that the merged proxy is less effective on target objectives that are more sensitive to small variations in the model's parameters. Importantly, this verifies only on a smaller subset of experimental settings.

## 4.2 Larger model size

We now assess the performance of merged proxies with larger models. To do so, we perform the most realistic (and complex) experiment with $K = 4$ domains, and use the merged proxies to select mixtures for Qwen2-VL-7B and Intern3.5-VL-8B. Results are reported in table 3, showing similar findings to those observed with 2B models. Merged proxies rank above median mixture performance in 11/14 and 9/14 specialist cases for Qwen2-VL-7B and Intern3.5-VL-8B, and yield mixtures within $-1\%$ gap from exact grid search in 11/14 and 10/14 cases, respectively. Yet again, we observe less sensitivity to mixture selection for both models when maximizing average performance. This is more pronounced in Intern3.5-VL-8B, hence we hypothesize this stems from the large fraction of instruction-tuning data already present in the Intern3.5-VL model family, which yields base models with broad knowledge and good instruction-following abilities already.

Table 3: Downstream accuracy of different mixtures for larger models (Qwen2-VL-7B and Intern3.5-VL-8B). Naming follows table 2.

| | | Target Benchmarks | | | | | | | | | | | | | |
| Model | Metric | GQA | OK-VQA | VQAv2 | VizWiz | DocVQA | InfoVQA | TextVQA | OCRBench | CVB-Count | Pope | ChartQA | MME | VMCBench | MMStar | Average Performance |
|---|---|---|---|---|---|---|---|---|---|---|---|---|---|---|---|---|
| Qwen2-VL-7B | Uniform | 60.9 | 51.2 | 76.5 | 63.8 | 93.3 | 72.1 | 81.6 | 79.8 | 61.7 | 88.0 | 81.3 | 79.8 | 78.3 | 52.0 | 72.9 |
| | Median | 61.0 | 46.9 | 77.2 | 67.2 | 93.1 | 72.0 | 81.0 | 79.6 | 60.6 | 88.1 | 81.3 | 79.2 | 77.8 | 52.3 | 72.6 |
| | Selected | 62.0 | 46.6 | 77.8 | 68.5 | 93.6 | 72.2 | 81.9 | 80.0 | 58.6 | 88.2 | 81.9 | 79.7 | 77.5 | 52.3 | 73.2 |
| | Best | 62.0 | 51.9 | 78.0 | 69.5 | 93.6 | 73.2 | 82.5 | 81.0 | 64.0 | 88.6 | 82.0 | 80.5 | 78.5 | 53.7 | 73.2 |
| Intern3.5-VL-8B | Uniform | 60.5 | 36.5 | 76.2 | 64.5 | 91.6 | 71.3 | 78.7 | 78.5 | 67.0 | 89.2 | 84.7 | 81.8 | 79.0 | 58.8 | 72.7 |
| | Median | 60.5 | 36.8 | 75.9 | 65.0 | 91.5 | 71.2 | 78.5 | 78.2 | 67.4 | 89.3 | 84.8 | 81.3 | 79.2 | 58.8 | 72.7 |
| | Selected | 61.3 | 37.5 | 76.4 | 65.2 | 91.6 | 72.5 | 78.8 | 78.5 | 66.6 | 89.2 | 84.8 | 80.7 | 78.9 | 58.3 | 72.8 |
| | Best | 61.3 | 42.9 | 76.4 | 66.1 | 91.9 | 72.5 | 79.1 | 78.6 | 69.4 | 89.6 | 85.5 | 83.1 | 79.8 | 59.5 | 73.0 |

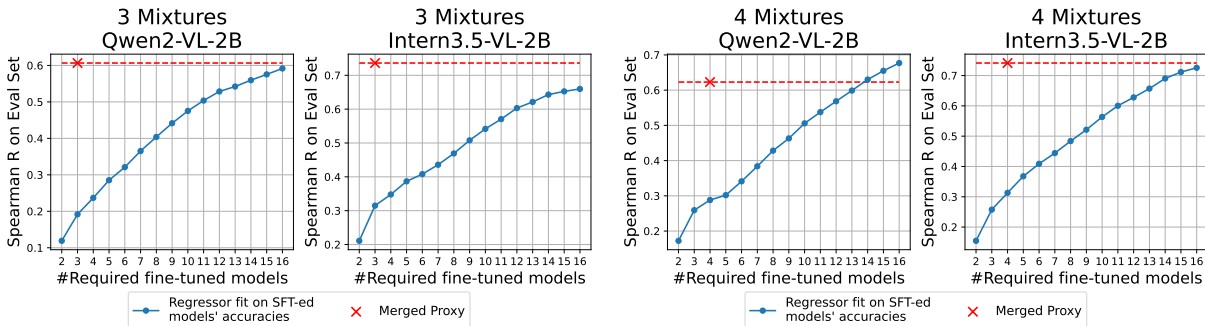

Figure 2: Spearman rank correlation between regressor-predicted and ground-truth accuracies, as a function of the number of training runs used to fit the regressor. The merged proxy requires only the $K$ domain-experts.

## 4.3 Merged Proxies vs Regression-based DMO

Established approaches for DMO resort to various forms of *regression*. For example, Shukor et al. (2025) fit power laws to regress a validation loss from domain weights $\mathbf{w}$, parameter count, and training tokens, while RegMix (Liu et al., 2025) fits a linear regression model to predict a validation loss from domain weights $\mathbf{w}$ only. Both approaches require sampling an initial population of training runs, and the more runs, the better. We now compare these regression-based approaches against merged proxies in the context of multimodal SFT.

**Setup.** We fit supervised regression models to predict the average accuracy on the 14 benchmarks given a population of $T$ training runs and evaluate how well they can *rank* mixtures in a held-out evaluation set with $n$ elements. We report details about the choice of the regressor in section C.5. We re-utilize the populations of training runs introduced above for both $K=\{3,4\}$ domains, to which we add expert models and 8 additional mixtures for a total of 21+3+8=32 and 20+4+8=32 mixtures, respectively. We fix the evaluation set size to $n = 16$ mixtures, and fit regression models on an increasing number of training runs $T = 2, ..., 24 - n$. Once fit, we use regression models to rank the mixtures in the evaluation set. We randomize this procedure for 1000 trials, reporting the expected values in figure 2. Recall that merged proxies only require $K$ training runs (one per domain).

**Observation #1: Merged Proxies are significantly more efficient than Regressors.** For Qwen2-VL-2B, both with 3-domains and 4-domains mixtures, the estimates provided by the regressor lag far behind those of the merged proxies, up until about 16 or 14 training runs have been collected. Similarly, for Intern3.5-VL-2B with 4-domains, the regressor requires about 16 training runs to match the merged proxy

performance. This means that merged proxies are approximately $4\times$ *more efficient* than regressors, as they require only a fraction of training runs to provide identical reliability.

**Observation #2: Regressors *may* never reach the performance of Merged Proxies.** For Intern3.5-VL-2B with 3-domains, we observe that regressors show signs of saturation while still lagging far behind merged proxies. This suggests that regressors may be more sensitive to the shape of the optimization landscape than merged proxies, and, importantly, they may require a very large number of training runs to match or surpass them.

While these observations might not be fully conclusive due to the modest population size, we hope that these results encourage the community to dig deeper into this matter.

## 5 Analysis and Intuitions

In this section, we provide intuition for *why* merged proxies serve as effective surrogates for mixture-finetuned models. We first derive a theoretical justification showing that, under local convexity assumptions in the loss landscape, the optimal mixture-trained weights are themselves linear combinations of expert weights. We then empirically verify these assumptions on our SFT data, and show that mixture-trained models are indeed aligned with merged proxies in parameter space, supporting the use of simple linear merging.

### 5.1 Second-order approximations lead to linear merging

We formally justify why a linear combination of experts $\sum w_i \boldsymbol{\theta}_i$ serves as a valid proxy for the mixture-trained model $\boldsymbol{\theta}_{\mathbf{w}}^*$ by analyzing the geometry of the loss landscape. The core assumption underlying our derivations is that losses remain approximately convex in a neighbourhood containing the domain experts. Although this may appear counterintuitive, as one would expect the loss landscape to be highly complex, with intricate hills and valleys, it is actually supported by established optimization literature. Specifically, the phenomenon of *linear mode connectivity* (LMC) (Frankle et al., 2020) shows that models fine-tuned from the same pre-trained base checkpoint reside in a shared low-loss basin, with no high-loss barriers separating them, therefore allowing the local landscape to be reliably approximated via second-order Taylor expansion.

Let $\ell(\theta; x)$ denote the loss on a single training sample. As described in section 3, a mixture $\mathcal{D}_{\mathbf{w}}$ is constructed by a process where each sample is selected by first sampling a domain index $z \sim \text{Categorical}(\mathbf{w})$, and then sampling $x \sim \mathcal{D}_z$. Therefore, the expected loss on the mixture is

$$\mathcal{L}(\theta, \mathcal{D}_{\mathbf{w}}) = \mathbb{E}_{z \sim \mathbf{w}} \mathbb{E}_{x \sim \mathcal{D}_z} \left[\ell(\theta; x)\right] = \sum_{i=1}^{K} w_i \mathbb{E}_{x \sim \mathcal{D}_i} \left[\ell(\theta; x)\right] = \sum_{i=1}^{K} w_i \mathcal{L}(\theta, \mathcal{D}_i).$$

For a finite mixture dataset of size $N$, the empirical loss is an unbiased estimate of this quantity, with approximately $w_i N$ samples per domain. Thus, in expectation, the loss on a mixture $\mathcal{D}_{\mathbf{w}}$ decomposes as

$$\mathcal{L}(\boldsymbol{\theta}, \mathcal{D}_{\mathbf{w}}) = \sum_{i=1}^{K} w_i \mathcal{L}(\boldsymbol{\theta}, \mathcal{D}_i). \tag{7}$$

Since the domain-experts are fine-tuned versions of the same base model, we can approximate the loss on each domain $\mathcal{D}_i$ using a second-order Taylor expansion around the corresponding expert $\boldsymbol{\theta}_i$, which reads

$$\mathcal{L}(\boldsymbol{\theta}, \mathcal{D}_i) \approx l_i + \frac{1}{2}(\boldsymbol{\theta} - \boldsymbol{\theta}_i)^{\top} \mathcal{H}_i (\boldsymbol{\theta} - \boldsymbol{\theta}_i), \tag{8}$$

where $l_i = \mathcal{L}(\boldsymbol{\theta}_i, \mathcal{D}_i)$ and $\mathcal{H}_i$ is the Hessian of the loss computed in $\boldsymbol{\theta}_i$. The first-order term is absent because the expert $\boldsymbol{\theta}_i$ is a minimiser of $\mathcal{L}(\boldsymbol{\theta}, \mathcal{D}_i)$, so the gradient is zero at this point. Consequently, we can write equation 7, the loss on the mixture $\mathcal{D}_{\mathbf{w}}$, by substituting equation 8 as

$$\mathcal{L}(\boldsymbol{\theta}, \mathcal{D}_{\mathbf{w}}) \approx \sum_{i} w_i [l_i + \frac{1}{2}(\boldsymbol{\theta} - \boldsymbol{\theta}_i)^{\top} \mathcal{H}_i (\boldsymbol{\theta} - \boldsymbol{\theta}_i)]. \tag{9}$$

The minimizer of this quadratic form has a closed-form solution

$$\boldsymbol{\theta}_{\mathbf{w}}^* = \left(\sum_j w_j \mathcal{H}_j\right)^{-1} \sum_i w_i \mathcal{H}_i \boldsymbol{\theta}_i, \tag{10}$$

which means that, under local convexity assumptions around experts, the optimal fine-tuned weights $\boldsymbol{\theta}_{\mathbf{w}}^*$ are themselves a linear combination of the expert models $\boldsymbol{\theta}_i$ with matrix coefficients. In the special case of uniform Hessians, i.e. $\mathcal{H}_i \approx \mathcal{H}$, $\forall i$, equation 10 simplifies to the linear combination of experts $\boldsymbol{\theta}_{\mathbf{w}}^* = \sum_i w_i \boldsymbol{\theta}_i$.

We note that equation 10 is closely related to existing curvature-aware merging methods, such as Fisher merging (Matena & Raffel, 2022), and that, from a Bayesian perspective, linear merging follows from approximating each expert's local posterior with an isotropic Gaussian of equal precision.

In practice, computing the full exact Hessians $\mathcal{H}_i \in \mathbb{R}^{|\boldsymbol{\theta}_i| \times |\boldsymbol{\theta}_i|}$ is not computationally feasible for modern MLLMs. While there exist techniques for estimating curvature information (Shen et al., 2024), they introduce additional computational and engineering costs. Since DMO does not require the optimal parameter-space approximation, but only a well-correlated surrogate, the uniform-Hessian simplification is a natural and principled starting point. We therefore adopt this simpler solution and leave the exploration of alternatives to future work.

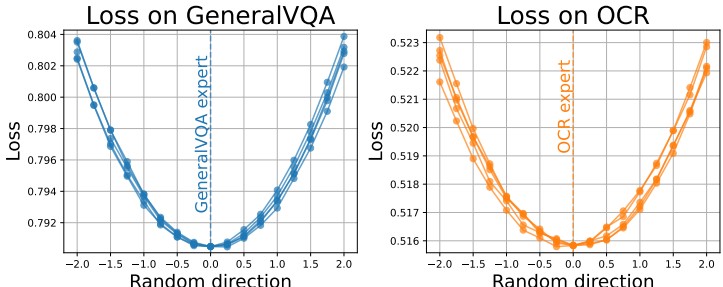

Figure 3: Loss functions in the neighbourhood of expert models, along 5 random directions. In a neighbourhood of their minimum, the loss functions remain convex.

## 5.2 Visualizing the loss function

We previously observed that the local-convexity assumption is supported by the fact that we are operating in a linear mode connectivity regime. We now visualize the loss function in the neighbourhood of the two expert models $\boldsymbol{\theta}_{\mathrm{general}}$, $\boldsymbol{\theta}_{\mathrm{ocr}}$ to empirically verify the approximation in equation 8. Precisely, for a given domain-expert model, we compute the loss values along five random directions $\boldsymbol{\delta}_j$ sampled from a normal distribution. To set a meaningful scale, we rescale the sampled directions to the distance between the two experts: $\boldsymbol{\delta} \leftarrow \frac{\boldsymbol{\delta}}{||\boldsymbol{\delta}||} ||\boldsymbol{\theta}_1 - \boldsymbol{\theta}_2||$ as in Li et al. (2018). We then evaluate the loss $\mathcal{L}(\mathcal{D}_i)$ in $\boldsymbol{\theta}_i + \alpha \boldsymbol{\delta}$ for evenly spaced alphas in $[-2, 2]$. Figure 3 visualizes the loss for General and OCR, which appears to satisfy the convexity assumption in its neighbourhood.

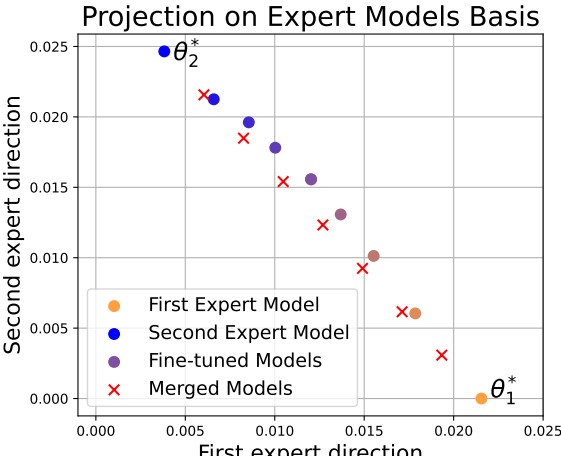

Figure 4: Projections of mixture-finetuned checkpoints are aligned between the two experts, suggesting they can be approximated by linearly merging the experts' parameters.

## 5.3 Parameter-space arrangement

To build intuition on how the finetuned models $\boldsymbol{\theta}_{\mathbf{w}}^*$ are arranged relatively to the expert models $\boldsymbol{\theta}_i$ in the parameters' space, we project models fine-tuned on

General + OCR data onto an orthonormal base centered in the base model and spanning the plane containing the two expert models $\boldsymbol{\theta}_{\mathrm{general}}$ and $\boldsymbol{\theta}_{\mathrm{ocr}}$ (we restrict to 2 domains for the sake of easier visualization). The projection is depicted in figure 4 for the grid of 7 mixtures. We can notice how the fine-tuned models are close to being organized along the straight line connecting the two specialized models, supporting the use of

linear merging. Additional visualizations supporting the assumptions underlying our analysis are reported in appendix C.6.

## 6 Conclusion

We investigate linear model merging as a practical surrogate for Data Mixture Optimization in supervised fine-tuning of multimodal large language models. By training one expert per domain and linearly merging their parameters, we obtain proxy models whose rankings are consistent with fully fine-tuned models. These merged proxies are largely independent of the optimization target (*specialist* vs. *generalist*) and remain robust as the number of domains increases. We also provide theoretical intuitions and empirical evidence explaining why linear combinations of experts are suitable surrogates for DMO, and show that merged proxies are far more sample-efficient than regression-based baselines. Overall, linear merging enables exploration of candidate mixtures using only one training run per domain, substantially reducing DMO cost.

**Limitations and Future Work**   In this work, we showed that a simple linear model merging can serve as an effective proxy for Data Mixture Optimization. Our primary goal was to establish the viability and empirical effectiveness of this strategy, settling the ground for multiple future research directions.

First, while we focused on linear model merging due to its simplicity and efficiency, merging strategies tailored specifically for DMO could yield further improvements. In particular, our analysis in section 5 suggests that incorporating second-order signals, such as the Fisher information matrix (Matena & Raffel, 2022), could produce more faithful approximations and potentially improve proxy quality.

Second, we currently treat the mixture search space as a finite grid rather than a continuous space. Future work could explore hybrid strategies that combine the merged proxy with continuous optimization techniques. In this sense, a possible solution is fitting a regressor on the performance of the merged proxies, which are far cheaper to collect than full fine-tuning runs, to enable the smooth interpolation of the objective over the mixture simplex. Alternatively, one could use the quadratic approximation surrogate proposed in (Li et al., 2025b) to efficiently optimize model merging coefficients. Another interesting approach could be to directly optimize the merging coefficients, for instance via gradient descent if the performance measure $f$ is differentiable, or through derivative-free methods such as genetic algorithms. In either case, the low cost of proxy evaluation permits a substantially deeper search than is feasible with standard mixture training.

Third, our problem setting assumes that the mixture weights remain fixed throughout SFT. This matches common practice (Tong et al., 2024; Deitke et al., 2025) and provides a controlled setting for evaluating whether merged models can serve as proxies for mixture-trained models. However, allowing for dynamic mixture weights varying over training enlarges the optimization space and may outperform the best static mixture (Zhu et al., 2025; Albalak et al., 2023). Extending the proposed merging-based method to such setting is therefore a valuable direction for future work. For example, one could decompose training into stages and use the merged proxies to choose the mixture for each stage. Alternatively, in a complementary approach, the selected static mixture can be used as a strong initialization for dynamic methods.

Finally, for the sake of wide and controllable analyses, we focused on two model families at two different scales, and a fixed data budget. It would be valuable to study how proxy quality varies with data budget by fitting scaling laws, which could reveal whether the approach generalizes across scales and remains effective in large-scale regimes where training costs are especially prohibitive.

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

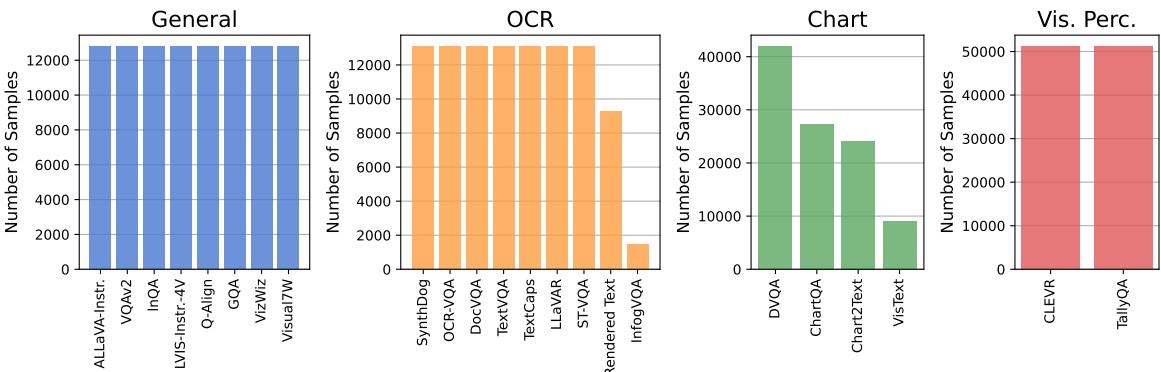

Figure 5: Composition of the domain-specific datasets used in our experiments.

# A   Broader Impact and Reproducibility Statements

**Broader Impact Statement**

The primary goal of our work is to render the expensive problem of Data Mixture Optimization more computationally accessible by using surrogate proxy models based on model merging. This implies that, when successful, such proxies could reduce both the environmental impact and the economic cost associated with optimizing Machine Learning models.

This work adds to the tools available to MLLM practitioners, and we do not foresee direct negative societal impacts specific to the proposed methodology. However, as with any technique that facilitates the development of more capable MLLMs, the broader consequences depend heavily on how the resulting models are used. The potential risks are not unique to our method and are shared with the broader landscape of MLLM research.

We finally note that our method does not introduce new data or model biases beyond those already present in the considered training datasets and base models. The choice of data sources and domain categories used during fine-tuning remains a responsibility of the practitioner, and we encourage careful curation of training data to mitigate the propagation of harmful biases into deployed systems.

**Reproducibility Statement**

The code and trained model checkpoints will be publicly released. The released checkpoints comprise over 150 models spanning four architectures: Qwen2-VL-2B, Qwen2-VL-7B, Intern3.5-VL-2B, and Intern3.5-VL-8B. These include both domain-expert checkpoints and mixture-finetuned models for different grids of mixtures. We hope that the availability of these resources will facilitate future research on DMO and support the development of new strategies building on the merging-based proxy introduced in this work.

# B   SFT Data & Benchmarks

In this section, we provide additional details about the training datasets and evaluation benchmarks used in our experiments.

**SFT data.**

In our experiments, we train models on data mixtures combining samples from four domain-specific datasets: General Multimodal Understanding, Optical Character Recognition (OCR), Visual Perception & Counting, and Charts Understanding. For each domain, we construct a dataset of $N = 102400$ (multiple of 128, the batch size) data points sampled uniformly and without repetitions from a collection of data sources, as described in table 1. Figure 5 reports the distribution over the data sources within each domain.

**Downstream tasks.** We evaluate our models on a suite of 14 benchmarks, evaluating a wide range of downstream tasks. For each of these, we now provide the split we tested on, the number of samples, and a brief description of their content and evaluated capabilities.

- **GQA** (Hudson & Manning, 2019) (testdev split, 12578 samples) A large-scale Visual Question Answering benchmark focused on real-world scene understanding with compositional reasoning. Questions are grounded in scene graphs, enabling the evaluation of relational, spatial, and logical reasoning.

- **OK-VQA** (Marino et al., 2019) (validation split, 5046 samples) An open-ended VQA dataset that requires external knowledge not directly observable in the image. It evaluates a model's ability to combine visual perception with commonsense and factual world knowledge.

- **VQAv2** (Goyal et al., 2017) (lite split, 500 samples) A widely used VQA benchmark containing balanced question–answer pairs to reduce language priors. It measures general visual understanding, object recognition, attributes, counting, and commonsense reasoning.

- **VizWiz** (Gurari et al., 2018) (validation split, 4319 samples) A VQA dataset collected from blind or low-vision users. Images are often of poor quality (blurred, poorly framed), making it a benchmark for robustness to real-world noise.

- **DocVQA** (Mathew et al., 2021) (validation split, 5349 samples) A document visual question answering benchmark requiring understanding of scanned documents such as forms and reports. It evaluates document layout comprehension, text localization, and reading-based reasoning.

- **InfoVQA** (Mathew et al., 2022) (validation split, 2801 samples) This benchmark focuses on question answering over infographics that combine text, charts, and visual elements. It requires multimodal reasoning across layout structure, textual content, and graphical components.

- **TextVQA** (Singh et al., 2019) (validation split, 5000 samples) A VQA benchmark emphasizing text understanding in real-world scenes. Many questions require reading text in natural images and, at the same time, integrating OCR outputs with visual context.

- **OCRBench** (Liu et al., 2024) (test split, 1000 samples) A comprehensive evaluation suite for OCR-centric multimodal models, covering text recognition, grounding, and reasoning across diverse real-world scenarios and document types.

- **CVBench2D (Counting)** (Tong et al., 2024) (test split, 788 samples) We use the *Counting* subtask, which evaluates fine-grained object counting in complex scenes. It requires precise object localization and reasoning on the numerosity of objects.

- **Pope** (Li et al., 2023) (test split, 9000 samples) A benchmark for evaluating object hallucination in vision–language models. We consider it a visual perception benchmark, since it measures whether models incorrectly assert the presence of objects not supported by the image.

- **ChartQA** (Masry et al., 2022) (test split, 2500 samples) A benchmark for question answering over charts and plots. It tests numerical reasoning and the ability to extract structured data from visualized statistics.

- **MME** (Fu et al., 2025) (test split, 2374 samples) A comprehensive multimodal evaluation benchmark designed to evaluate perception, reasoning, and knowledge capabilities of multimodal foundation models across diverse task categories.

- **VMC-Bench** (Zhang et al., 2025b) (dev split, 1000 samples) This benchmark evaluates a model's competence across multiple cognitive dimensions, such as perception, reasoning, and cross-modal consistency.

- **MMStar** (Chen et al., 2024b) (validation split, 1500 samples) This is a comprehensive benchmark designed to assess structured, step-by-step reasoning ability in vision–language models.

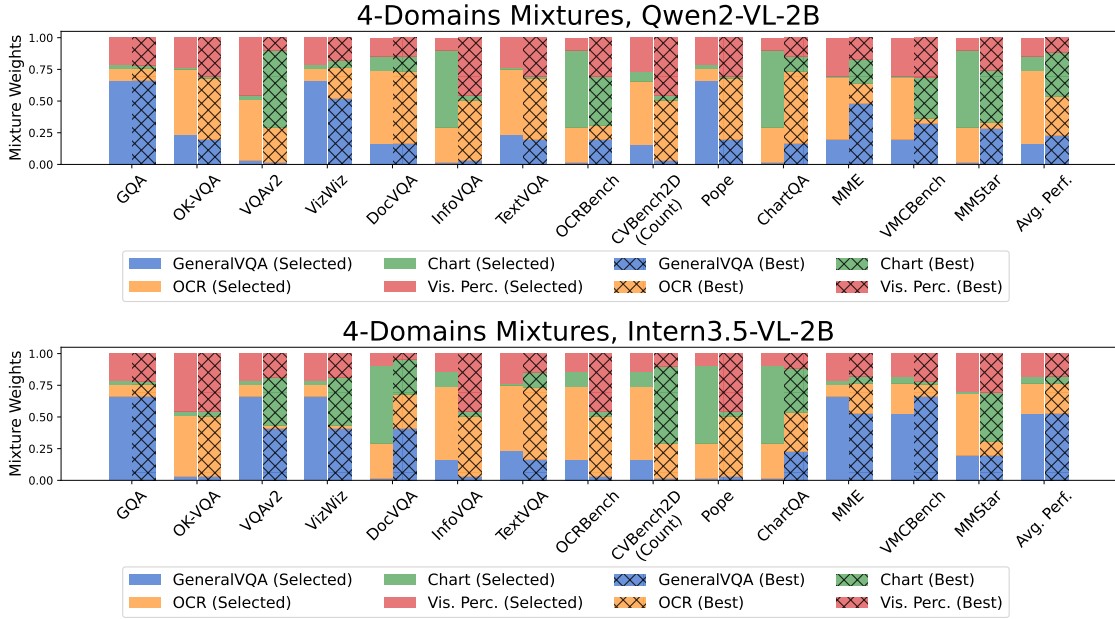

Figure 6: Comparison of optimal data mixtures and data mixtures selected by the merged proxy.

## C  Additional Experiments

In this section, we report additional experimental results that complement and extend those in the main document. We first analyze the optimal mixtures under diverse target objectives, and we compare them with those selected by our merged proxy (appendix C.1). We then observe that merging experts fine-tuned at a fraction of the target data budget is sufficient to obtain strong proxies (section C.2). Next, we evaluate our proposed merged proxy at different data budgets (appendix C.3), and on full-finetuning (appendix C.4). We report additional details about the regression-based comparison (appendix C.5). Finally, we perform a preliminary investigation of the effect of optiming over the full continuous simplex of mixing weights.

### C.1  Optimal mixtures

In figure 6, we report, for each of the 14 benchmarks and in the setting of 4-domain mixtures, both the best-performing mixture found via exact grid search and the mixture selected by the merged proxies. A notable observation is that the resulting optimal compositions often diverge from what one might expect based on human intuition about dataset–task alignment.

For instance, for both Qwen2-VL and Intern3.5-VL, performance on OK-VQA is maximized by mixtures that allocate a substantial fraction of OCR and Visual Perception data. Importantly, these allocations are consistently identified both by the merged proxies and by exhaustive search. This outcome is somewhat counterintuitive, as OK-VQA would typically be categorized as a General benchmark, and one might therefore expect mixtures dominated by general-purpose data. A related phenomenon is observed for VQAv2, another benchmark commonly regarded as General. In this case, Qwen2-VL benefits most from mixtures emphasizing OCR and Charts data, whereas Intern3.5-VL does not exhibit the same preference.

These discrepancies indicate that the optimal supervised fine-tuning (SFT) mixture is strongly influenced by the prior knowledge and inductive biases of the underlying base model. Consequently, mixture compositions that are optimal for one architecture cannot be assumed to transfer to another, even when targeting the same downstream task. Although intuitive, this result highlights that mixture optimization for SFT is inherently model-dependent. Importantly, these instances of positive cross-domain transfer are effectively captured by the merged proxies.

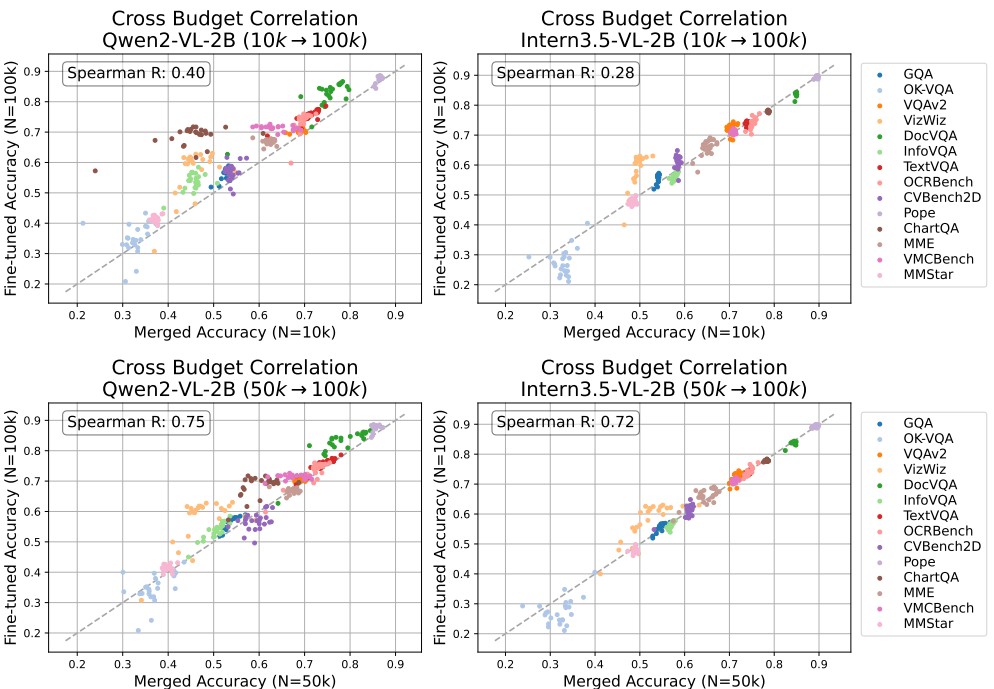

Figure 7: Cross-data budget correlation plots. Results are shown for Qwen2-VL-2B and Intern3.5-VL-2B models for 4-domains mixtures.

Overall, these findings demonstrate that optimal data mixtures can be unintuitive, and highlight the fact that automated mixture optimization rather than reliance on heuristic or human-designed compositions is to be preferred.

## C.2   Cross Budget Correlation

We investigate whether merged proxies constructed from experts trained with a smaller data budget can still reliably rank mixtures defined at a larger target budget. Let the target budget be $N_{\text{target}} = 100k$ samples. Rather than training experts on the full budget, we build proxy models by merging experts trained with reduced budgets $N_{\text{proxy}} = 10k, 50k$. We then measure how well proxy-based scores correlate with the true performance of models fine-tuned at $N_{\text{target}}$ across the grid of 4-domain mixtures.

Figure 7 reports the correlation between proxy scores and target accuracies for Qwen2-VL-2B and Intern3.5-VL-2B. Proxies derived from the smallest budget ($N_{\text{proxy}} = 10k$) exhibit weak correlation with target performance, suggesting that experts trained with very limited data fail to capture sufficiently reliable domain-specific signals. In contrast, proxies constructed from half-budget experts ($N_{\text{proxy}} = 50k$) achieve strong correlation with the target scores for both models.

Overall, these results indicate that merged proxies do not require experts trained at the full optimization budget. Instead, training experts on a fraction of the data can already provide a faithful ranking signal over mixtures, enabling a further reduction in the total computational cost of the merging-based DMO framework. Furthermore, this finding opens the possibility of leveraging merged proxy predictions in scaling laws fitted on smaller data budgets, which presents an interesting direction for future research.

## C.3   Different Data Budgets

We investigate the behavior of merged proxies under different data budgets. To this end, we repeat the main experiment from Section 4.1 by sampling from the 4-domain mixture space with progressively larger budgets: $N = 10240, 51200, 102400$. Table 4 reports results for both Qwen2-VL-2B and Intern3.5-VL-2B.

Table 4: Downstream accuracies of models fine-tuned on 4-domains data mixtures. We compare the performance of the mixture selected by our merged proxy (**Selected**) against the **Uniform**, **Median** and **Best** mixtures on an increasing number of data samples allocated to SFT ($N =$10k, 50k, 100k samples). Naming and formatting follow table 2.

| Target Benchmarks | $N = 10k$ SAMPLES | | | | $N = 50k$ SAMPLES | | | | $N = 100k$ SAMPLES | | | |
|---|---|---|---|---|---|---|---|---|---|---|---|---|
| | Uniform | Median | Selected | Best | Uniform | Median | Selected | Best | Uniform | Median | Selected | Best |
| *Qwen2-VL-2B* | | | | | | | | | | | | |
| GQA | 53.0 | 53.1 | 53.8 | 54.1 | 55.7 | 55.6 | 56.9 | 56.9 | 56.9 | 57.5 | 58.4 | 58.4 |
| OK-VQA | 33.0 | 31.0 | 35.7 | 35.7 | 33.7 | 27.1 | 38.7 | 38.7 | 35.8 | 34.6 | 39.0 | 40.7 |
| VQAv2 | 68.6 | 68.2 | 69.2 | 69.9 | 69.7 | 69.5 | 69.4 | 70.7 | 70.5 | 70.5 | 70.9 | 71.6 |
| VizWiz | 49.9 | 51.6 | 55.3 | 55.3 | 59.0 | 58.8 | 57.6 | 62.6 | 62.0 | 61.0 | 58.5 | 63.0 |
| DocVQA | 78.8 | 76.7 | 80.4 | 80.5 | 83.1 | 81.9 | 84.6 | 85.2 | 85.1 | 84.1 | 86.6 | 86.6 |
| InfoVQA | 48.0 | 47.3 | 50.4 | 50.4 | 54.0 | 53.7 | 54.6 | 55.8 | 54.1 | 54.6 | 56.1 | 58.4 |
| TextVQA | 73.0 | 71.8 | 74.1 | 74.1 | 75.5 | 74.5 | 76.2 | 77.4 | 76.6 | 75.7 | 76.4 | 77.5 |
| OCRBench | 70.3 | 70.2 | 71.5 | 72.0 | 74.0 | 73.5 | 74.6 | 75.1 | 75.2 | 74.7 | 75.4 | 75.6 |
| CVBench2D-Count | 52.0 | 52.2 | 54.9 | 54.9 | 57.1 | 57.4 | 58.6 | 60.2 | 57.9 | 56.6 | 56.5 | 61.3 |
| Pope | 86.8 | 86.6 | 87.0 | 87.1 | 87.8 | 87.3 | 87.4 | 87.9 | 88.3 | 88.2 | 88.5 | 88.6 |
| ChartQA | 52.0 | 53.1 | 58.0 | 58.0 | 69.0 | 68.0 | 70.4 | 70.4 | 70.3 | 70.0 | 71.6 | 71.7 |
| MME | 62.3 | 62.8 | 62.8 | 64.4 | 66.2 | 65.9 | 66.5 | 67.2 | 67.7 | 67.0 | 66.9 | 67.8 |
| VMCBench | 64.2 | 64.5 | 69.6 | 69.6 | 70.8 | 70.4 | 70.7 | 71.3 | 70.9 | 71.7 | 70.8 | 72.6 |
| MMStar | 39.0 | 38.3 | 39.2 | 39.2 | 40.6 | 40.2 | 40.0 | 42.0 | 41.4 | 41.3 | 41.7 | 42.7 |
| **Average Performance** | 59.4 | 59.1 | 60.0 | 60.0 | 64.0 | 63.1 | 63.8 | 63.9 | 65.2 | 64.8 | 65.3 | 65.5 |
| *Intern3.5-VL-2B* | | | | | | | | | | | | |
| GQA | 54.4 | 54.5 | 54.6 | 54.8 | 55.4 | 55.3 | 56.2 | 56.2 | 56.2 | 56.4 | 56.9 | 56.9 |
| OK-VQA | 28.5 | 28.3 | 33.3 | 33.4 | 24.5 | 22.0 | 33.0 | 33.0 | 25.2 | 26.3 | 32.2 | 32.2 |
| VQAv2 | 70.4 | 70.6 | 71.9 | 71.9 | 72.8 | 72.5 | 73.3 | 73.3 | 73.6 | 73.2 | 73.9 | 74.5 |
| VizWiz | 51.6 | 52.5 | 52.9 | 53.0 | 57.0 | 60.2 | 62.0 | 62.0 | 58.9 | 61.2 | 62.6 | 63.0 |
| DocVQA | 84.7 | 84.9 | 85.0 | 85.0 | 84.3 | 84.1 | 84.4 | 84.4 | 84.3 | 83.8 | 84.2 | 84.4 |
| InfoVQA | 58.5 | 58.3 | 59.0 | 59.0 | 56.9 | 56.5 | 57.2 | 57.5 | 56.3 | 56.4 | 56.8 | 57.9 |
| TextVQA | 74.0 | 73.8 | 73.7 | 74.2 | 74.2 | 74.2 | 74.4 | 74.6 | 74.3 | 74.0 | 74.7 | 74.7 |
| OCRBench | 75.6 | 75.3 | 76.0 | 76.2 | 74.6 | 74.4 | 75.6 | 75.7 | 75.0 | 73.9 | 75.2 | 76.6 |
| CVBench2D-Count | 58.6 | 58.8 | 58.1 | 59.4 | 60.7 | 59.9 | 60.9 | 62.4 | 60.5 | 61.2 | 62.6 | 64.8 |
| Pope | 89.8 | 89.4 | 89.4 | 89.6 | 89.7 | 89.6 | 89.6 | 90.1 | 89.3 | 89.5 | 89.8 | 89.9 |
| ChartQA | 78.6 | 78.7 | 78.6 | 78.9 | 78.4 | 78.2 | 78.2 | 78.8 | 78.2 | 77.9 | 78.0 | 78.4 |
| MME | 66.2 | 66.6 | 68.5 | 68.5 | 64.4 | 65.7 | 66.1 | 68.8 | 66.9 | 66.4 | 68.5 | 69.1 |
| VMCBench | 70.6 | 70.4 | 71.1 | 71.4 | 70.8 | 70.8 | 70.6 | 71.4 | 71.3 | 71.2 | 71.5 | 72.2 |
| MMStar | 46.7 | 46.9 | 48.5 | 48.6 | 46.5 | 47.0 | 47.3 | 47.7 | 47.7 | 47.4 | 47.3 | 48.5 |
| **Average Performance** | 64.9 | 64.9 | 65.1 | 65.1 | 65.0 | 64.9 | 65.5 | 65.5 | 65.5 | 65.5 | 66.0 | 66.0 |

We find that merged proxies remain effective even at lower data budgets, consistently selecting strong mixtures. In contrast, the same trend does not hold for the Uniform mixture. While its broad task coverage yields reasonably good performance at the largest budget (100k), it becomes a much weaker choice when the data is limited, particularly for Qwen2-VL-2B. For instance, at a budget of 10k examples, the Uniform mixture underperforms the best mixture (and the one selected by merged proxies) by $-6\%$ on ChartQA, $-5.4\%$ on VizWiz, and $-5.4\%$ on VMC-Bench. These results indicate that, especially in low-budget regimes, careful mixture selection is important, and merged proxies provide a reliable mechanism to achieve this.

## C.4 Full finetuning

In table 5, we report the performance of merged proxies compared to the Uniform, Median, and Best baselines for 4-domain mixtures, using *full* fine-tuning on Qwen2-VL-2B and Intern3.5-VL-2B. In this setting, we do not tune hyperparameters specifically for full fine-tuning; instead, we reuse exactly the same hyperparameters described in section 4. The only change with respect to previous experiments is that, rather than applying low-rank adaptation, we fine-tune all weights of the language decoder and multimodal adapter.

Despite this change in training regime, we continue to observe strong results. In particular, for both models, merged proxies consistently select near-optimal mixtures in the *generalist* scenario (see the **Average Performance** column), outperforming both Uniform and Median baselines and closely approaching the performance achieved by exact grid search.

We also observe slightly larger gaps between the best mixture found by grid search and the mixture selected by merged proxies, in both specialist and generalist scenarios. We hypothesize that this behavior is largely attributable to suboptimal hyperparameter choices for full fine-tuning. For example, a learning rate that is

Table 5: Downstream accuracy of different 4-domains mixtures for Qwen2-VL-2B and Intern3.5-VL-2B trained with **full fine-tuning**.

| Model | Metric | GQA | OK-VQA | VQAv2 | VizWiz | DocVQA | InfoVQA | TextVQA | OCRBench | CVB-Count | Pope | ChartQA | MME | VMCBench | MMStar | Average Performance |
|---|---|---|---|---|---|---|---|---|---|---|---|---|---|---|---|---|
| | | | | | | | | | | | | | | | | Target Benchmarks |
| Qwen2-VL-2B Full Fine-tuning | Uniform | 60.39 | 42.14 | 69.78 | 52.40 | 87.22 | 56.65 | 76.11 | 73.10 | 54.19 | 86.74 | 73.00 | 64.19 | 70.20 | 42.16 | 64.88 |
| | Median | 61.18 | 42.60 | 69.53 | 53.28 | 85.94 | 55.43 | 75.34 | 71.85 | 58.25 | 87.02 | 73.14 | 66.21 | 70.60 | 42.84 | 65.09 |
| | Selected | 61.86 | 42.07 | 71.10 | 55.97 | 87.26 | 57.70 | 77.73 | 72.90 | 54.57 | 86.73 | 74.88 | 69.23 | 70.40 | 40.59 | 65.61 |
| | Best | 62.41 | 46.02 | 71.62 | 59.22 | 88.47 | 58.83 | 78.82 | 73.80 | 66.50 | 88.46 | 74.88 | 69.87 | 72.70 | 46.37 | 66.67 |
| Intern3.5-VL-2B Full Fine-tuning | Uniform | 58.18 | 41.37 | 72.24 | 40.39 | 83.19 | 54.39 | 71.33 | 71.80 | 60.53 | 88.99 | 74.64 | 64.51 | 70.30 | 48.26 | 64.29 |
| | Median | 58.27 | 41.75 | 70.06 | 41.44 | 81.96 | 53.05 | 71.02 | 72.00 | 63.13 | 89.11 | 75.72 | 67.66 | 71.35 | 48.47 | 64.48 |
| | Selected | 60.14 | 41.74 | 72.22 | 44.16 | 83.71 | 54.53 | 71.82 | 73.90 | 61.42 | 88.99 | 76.56 | 61.49 | 71.20 | 49.06 | 65.01 |
| | Best | 60.14 | 43.75 | 72.22 | 47.27 | 84.11 | 56.36 | 73.98 | 75.00 | 68.15 | 90.08 | 76.88 | 71.61 | 72.70 | 49.70 | 66.01 |

too high when updating all weights may increase the distances between experts, which in turn can reduce the effectiveness of model merging and slightly degrade proxy reliability.

## C.5 Additional Details about Regression-based DMO

In section 4, we presented experiments comparing regression-based alternatives to the merged proxies. For the main body of the paper, we considered a quadratic Ridge regressor, as it achieved highest correlation with ground-truth performance among the regressors we evaluated. For completeness, figure 8 reports results for a linear Ridge regressor and a LightGBM regressor (Ke et al., 2017) in the 4-domain setting.

Both regressors substantially underperform the merged proxies despite requiring several additional training runs. These results further support the findings in section 4: regression-based approaches exhibit varying capacities to model mixture-performance relationships, and increasing the number of observed runs alone does not guarantee that the same effectiveness of merged proxies is reached. In contrast, merged proxies appear better suited in low-data regimes, proving a better alternative for mixture selection under a limited compute budget.

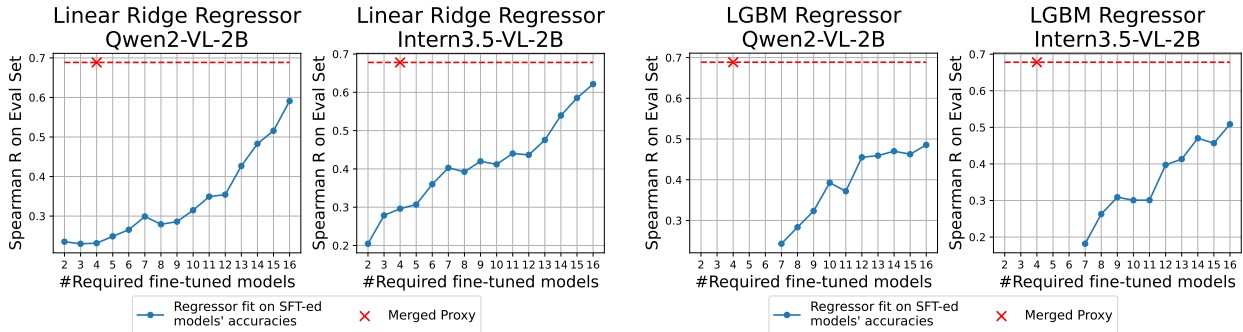

Figure 8: Spearman's R correlation coefficient of accuracies predicted by a linear ridge regressor (top) and a LightGBM regressor (bottom), fitted on an increasing number of data points. Each data point comes from a finetuned model, while the merged proxy requires only the $K$ expert models.

## C.6 Additional visualizations for Section 5 Analysis

In this section, we extend the visualizations presented in section 5. In figure 9, we visualize the domain-specific losses in a neighbourhood of their minima for all four domains considered in our experimental evaluation,

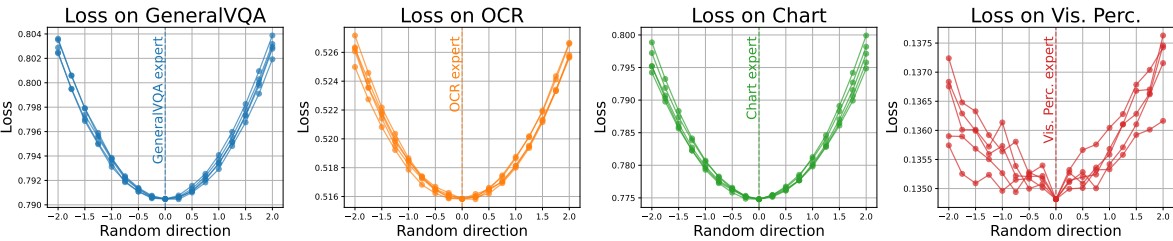

Figure 9: Loss functions in the neighbourhood of expert models, along 5 random directions. In a neighbourhood of their minimum, the loss functions remain convex.

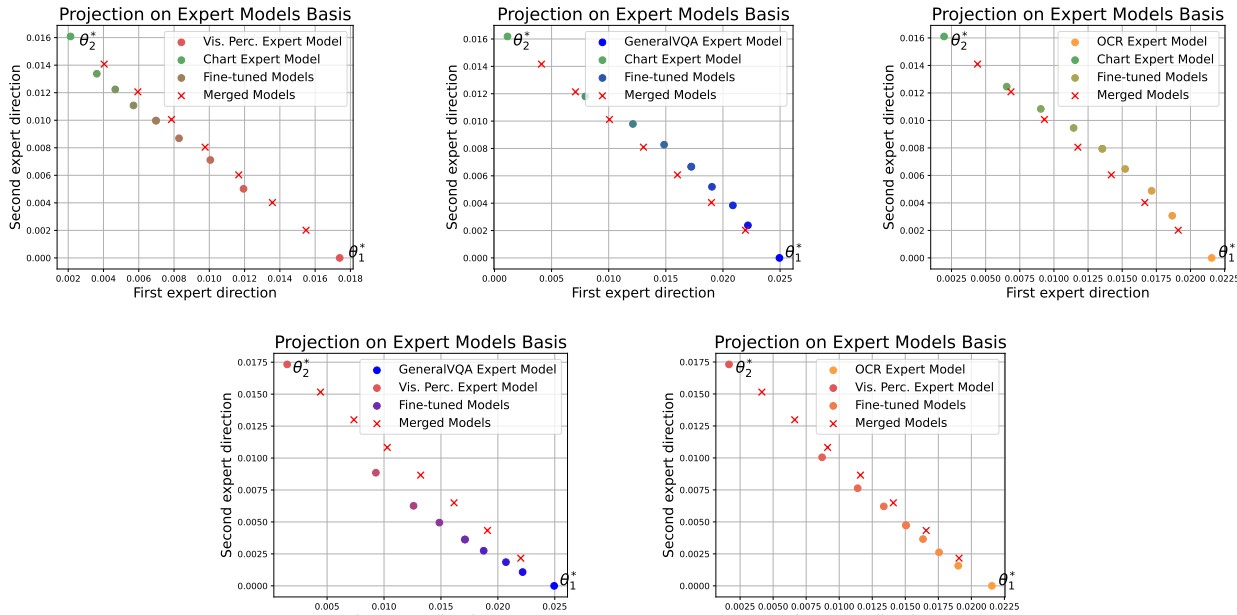

Figure 10: Projections of mixture-finetuned checkpoints are aligned between the two experts, suggesting they can be approximated by linearly merging the experts' parameters.

similarly to figure 3 in the main document. Overall, these new visualizations support the local convexity assumptions and further verify the observations in the main document. Only the loss on Visual Perception is less smooth compared to the others, but still displays convexity. Figure 4 in the main document visualizes the projections on the 2D plane spanned by the General VQA and OCR experts. Similarly, figure 10 reports the same visualization for the five other possible pairs of four domains. Also for these combinations, the mixture-finetune models are close to be aligned along the two experts, further supporting the use of linear merging as a proxy for DMO.

## C.7 Continuous optimization

We perform a preliminary experiment investigating an hybrid approach combining the merged proxy with continuous optimization techniques. We fit a regressor on the performance of the merged proxies, and then optimize the predicted merged scores over a finer grid of mixing weights. We perform this experiment in the setting with four domains. Results in table 6 suggest that optimizing over the continuous simplex may lead to significant improvents on specialist objectives, while it provides no visible improvements over the already strong Average Performance of the mixtures selected on the finite grid. While limited to a single setting, this preliminary analysis suggests that hybrid methods are a valauble direction for future research.

Table 6: Performance of mixtures selected by a regressor fitten on merged models performance.). Naming follows table 2.

| Model | Metric | GQA | OK-VQA | VQAv2 | VizWiz | DocVQA | InfoVQA | TextVQA | OCRBench | CVB-Count | Pope | ChartQA | MME | VMCBench | MMStar | Average Performance |
|---|---|---|---|---|---|---|---|---|---|---|---|---|---|---|---|---|
| | | | | | | | | | **Target Benchmarks** | | | | | | | |
| Qwen2-VL-2B | Uniform | 56.9 | 35.8 | 70.5 | 62.0 | 85.1 | 54.1 | 76.6 | 75.2 | 57.9 | 88.3 | 70.3 | 67.7 | 70.9 | 41.4 | 65.2 |
| | Median | 57.4 | 34.5 | 70.4 | 61.0 | 84.1 | 54.6 | 75.8 | 74.8 | 56.8 | 88.2 | 70.1 | 66.9 | 71.5 | 41.2 | 64.8 |
| | Selected | 58.4 | 39.0 | 70.9 | 58.5 | 86.6 | 56.1 | 76.4 | 75.4 | 56.5 | 88.5 | 71.6 | 66.9 | 70.8 | 41.7 | 65.3 |
| | Selected (reg.) | 58.9 | 43.3 | 71.3 | 57.1 | 86.6 | 56.1 | 78.6 | 75.8 | 62.7 | 87.9 | 69.6 | 67.0 | 71.8 | 43.0 | 65.3 |
| | Best | 58.4 | 40.7 | 71.6 | 63.0 | 86.6 | 58.4 | 77.5 | 76.0 | 61.3 | 88.6 | 71.7 | 67.8 | 72.6 | 42.7 | 65.5 |
| Intern3.5-VL-2B | Uniform | 56.2 | 25.2 | 73.6 | 58.9 | 84.3 | 56.3 | 74.3 | 75.0 | 60.5 | 89.3 | 78.2 | 66.9 | 71.3 | 47.7 | 65.5 |
| | Median | 56.4 | 26.3 | 73.2 | 61.0 | 83.8 | 56.3 | 73.9 | 74.0 | 61.2 | 89.6 | 77.9 | 66.3 | 71.2 | 47.4 | 65.5 |
| | Selected | 56.9 | 32.2 | 73.9 | 62.6 | 84.2 | 56.8 | 74.7 | 75.2 | 62.6 | 89.8 | 78.0 | 68.5 | 71.5 | 47.3 | 66.0 |
| | Selected (reg.) | 57.3 | 40.5 | 73.4 | 63.0 | 84.4 | 57.8 | 74.7 | 77.2 | 63.2 | 89.8 | 78.1 | 68.1 | 71.5 | 49.8 | 66.0 |
| | Best | 56.9 | 32.2 | 74.5 | 63.0 | 84.4 | 57.9 | 74.7 | 76.6 | 64.8 | 89.9 | 78.4 | 69.1 | 72.2 | 48.7 | 66.0 |

