# OpenReview forum: "Linear Model Merging Unlocks Simple and Scalable Multimodal Data Mixture Optimization"
_TMLR — Under review for TMLR_

### Review · Reviewer_obE2 · 2026-06-16

**Summary Of Contributions:**

## Summary of Contributions

On a high level, the paper tackles the problem of Data Mixture Optimization (DMO) for multimodal supervised fine-tuning (SFT), i.e., given a set of **K** domain-specific datasets (General, OCR, Visual Perception, Charts), how should one decide the mixture weights **w** to fine-tune on in order to get the best downstream performance? The naive approach would be to grid-search over candidate mixtures. This becomes increasingly costly as **K** increases.

The motivation of the study is to propose a "simple yet effective" proxy for the above problem. Instead of training a model for every candidate mixture, the paper proposes training only **K** single-domain "expert" models, and then linearly merging their weights according to the candidate mixture weights **w** to obtain a "merged proxy" model. The accuracy of this proxy model on the downstream benchmarks is then used to rank and select the best mixture, without ever training on the mixture itself.

In essence, the paper makes the following key contributions

1. Train one "expert" model per domain (**K** experts total), then linearly interpolate their weights according to a candidate mixture **w** to obtain a merged proxy model, whose downstream accuracy is used to rank candidate mixtures.
2. A theoretical justification (via a second-order Taylor expansion of the loss around each expert, under a local convexity / linear mode connectivity assumption) showing that the optimal mixture-trained weights reduce to a Hessian-weighted combination of experts, which simplifies to plain linear merging under a uniform-Hessian assumption.
3. Comprehensive empirical validation across two model families (Qwen2-VL and Intern3.5-VL), multiple scales (2B and 7B/8B parameters), both LoRA and full fine-tuning, and 14 multimodal benchmarks, comparing the merged proxy approach against grid-search optima and regression-based DMO baselines (RegMix-style).

The paper also draws insights into when DMO actually matters: in specialist settings (optimizing for a single benchmark), the proxy-selected mixture lands within roughly 1% of the grid-search optimum in most cases, while in generalist settings (averaged across benchmarks), the gap between median and best mixtures is small to begin with, so the proxy's advantage matters less there. Compared against regression-based DMO, the merged proxies (needing only **K** training runs) match or beat regressors fit on 15+ full fine-tuning runs, which is a meaningful efficiency win.

### Strengths

1. **Practicality and Efficiency**: The headline contribution, that a handful of expert training runs can stand in for an expensive grid search over mixtures, is genuinely useful and cheap to adopt. The comparison against regression-based DMO baselines (Table/Fig 2) is particularly compelling, since the merged proxy needs only **K** runs and still matches or beats regressors trained on 15-16+ full fine-tuning runs, even where regressors plateau below the proxy's correlation with 4x the compute for Intern3.5-VL.

2. **Breadth of Experimentation**: The paper backs up its claims with an extensive empirical sweep, two model families, multiple scales (2B and 7B/8B), LoRA as well as full fine-tuning, varying **K** (2/3/4 domains), and varying data budgets (10k/50k/100k), evaluated across 14 benchmarks. This breadth lends credibility to the central claim and makes the results more generalizable than a single-setting demonstration would.

3. **Honest Reporting of Nuance**: The paper is upfront about cases where the proxy underperforms (e.g. OK-VQA, VizWiz, and CVBench2D-Count for Qwen2-VL show high variance and larger gaps from the optimum), and is candid that DMO matters less in generalist settings since median and best mixtures are already close. This kind of transparency strengthens trust in the overall claims rather than overselling them.

4. **Reproducibility**: The promise of releasing 150+ checkpoints along with detailed hyperparameters and dataset composition tables is a strong practical contribution in its own right, and paves the way for follow-up work to build directly on this benchmark.

### Weaknesses

1. **Strength of the Core Correlation Claim**: The reported Spearman correlations between merged-proxy accuracy and actual mixture-trained accuracy range from 0.57 to 0.78, which is a moderate range rather than an exceptionally strong one. It's important to note that the downstream mixture-selection results (within ~1% of grid search in most cases) are the more convincing evidence, so the framing in the abstract around "high rank correlation" could be argued/contextualized better against these more moderate raw correlation numbers.

2. **Theory-to-Practice Gap**: The theoretical derivation (Eq. 10) naturally yields a Hessian-weighted combination of experts, and the simplification to plain linear merging relies on a uniform-Hessian assumption that is acknowledged but not rigorously justified beyond empirical plausibility (Figs 3-4), which themselves are only shown for the **K**=2 General+OCR case. Extending this empirical support to additional domain pairs and higher **K** would strengthen the theoretical grounding considerably.

3. **Search Space Limitation**: The paper restricts itself to a finite grid of mixtures rather than a continuous search space, and does not demonstrate any gradient-based or iterative refinement of mixture weights. This is acknowledged as a limitation by the authors, but addressing it, even partially, would significantly broaden the practical applicability of the method.

4. **Statistical Robustness of Comparisons**: The "Selected" results rely on only 3 seeds, and the regression-based comparison, while run over 100 trials, draws from a fairly small population of 24 total mixtures, something the authors themselves flag as not fully conclusive. Additional seeds or a larger underlying mixture population would help solidify these comparisons and add weight to the paper's claims.

### Minor Weakness
The independent/concurrent work by Wang et al. (2026), which also explores model merging as a DMO proxy but in the text-only LLM setting, is discussed as related work; the delta relative to this paper (the multimodal setting) could be stated slightly more explicitly upfront to preempt novelty questions during review.

**Audience:**

Yes

**Audience Explanation:**

The findings of this paper are relevant and practically useful to a broad section of TMLR's audience. On a high level, Data Mixture Optimization is not a niche problem — it is a core bottleneck faced by anyone fine-tuning large multimodal models on heterogeneous domain-specific datasets, which describes a significant and growing segment of the ML community. The paper directly addresses this bottleneck with a method that is simple to implement, requires no additional infrastructure beyond standard fine-tuning, and is backed by rigorous empirical validation.

The findings are of interest across at least three distinct communities within TMLR's readership

1. **Multimodal LLM researchers and practitioners** will find the benchmark and insights immediately actionable. The paper's result that a small set of domain expert models can stand in for an expensive mixture grid search has direct implications for how multimodal SFT pipelines are designed, and the release of 150+ checkpoints paves the way for follow-up work.

2. **Model merging researchers** will find the paper's framing of linear interpolation as a DMO proxy — rather than a performance target in itself — a fresh and well-motivated perspective. The theoretical analysis connecting merging to mixture optimization via Taylor expansion and Hessian geometry draws insights that are relevant beyond the multimodal setting.

3. **Researchers working on efficient training and hyperparameter optimization** will find the comparison against regression-based DMO baselines (RegMix-style) compelling. The result that K expert training runs match or outperform regressors fit on 15-16+ full fine-tuning runs has implications for how the broader community approaches proxy-based search in compute-constrained settings.

It is also worth noting that the paper's insight into *when* DMO matters, in that it matters significantly in *specialist* settings but less so in *generalist* ones. This observation is non-obvious and practically valuable finding in its own right. This kind of nuanced, well-supported conclusion is precisely what draws insights from benchmarking work into the broader research conversation.

**Broader Impact Concerns:**

The paper does not include a Broader Impact Statement. That said, based on the nature of the paper's contribution, this omission does not raise significant concerns. The contribution is methodological in nature as it proposes a compute-efficient proxy for Data Mixture Optimization (DMO) during supervised fine-tuning (SFT) of multimodal large language models (MLLMs) and does not introduce new capabilities that could directly enable harm.

That said, there are a few indirect considerations worth briefly noting, though none of them rise to the level of requiring a dedicated Broader Impact Statement in this reviewer's assessment.

1. **Efficiency enabling broader fine-tuning access**: By significantly reducing the compute cost of data mixture search, the method lowers the barrier to fine-tuning large multimodal models across diverse domains. This is broadly positive, as it makes high-quality fine-tuning more accessible to researchers and organizations with limited compute. It does, however, also make it easier to fine-tune models for specialized tasks at scale, which could include applications that require careful consideration (e.g., domain-specific generation in sensitive areas). This is a general concern with efficiency improvements in ML and is not specific to this paper.

2. **Model release**: The promise to release 150+ checkpoints is a positive reproducibility contribution. The authors may wish to consider briefly noting any intended usage guidelines for the released models, particularly given that fine-tuned multimodal models can be repurposed beyond their intended evaluation scope.

Overall, the broader impact of this work is positive and it contributes to making multimodal model fine-tuning more efficient and reproducible. The above concerns are minor and common to the broader category of efficiency-focused ML research. No changes are required on ethical grounds.

**Claims And Evidence:**

Yes

**Claims Explanation:**

The paper's central claim is that linearly merging domain-expert weights provides a reliable and compute-efficient proxy for ranking Data Mixture Optimization (DMO) candidates in multimodal SFT. This claim is thoroughly backed comprehensive and well-structured empirical body of evidence. The paper does not make bold, unsupported assertions, rather, it is careful to scope its claims and be upfront about where the evidence is weaker. That said, there are a few gaps worth flagging.

### Where the evidence is convincing

The most compelling evidence comes from the mixture-selection results rather than the raw correlation numbers. The merged proxy selects mixtures that land within roughly 1% of the grid-search optimum in the majority of specialist benchmark cases across both Qwen2-VL and Intern3.5-VL model families, at 2B and 7B/8B scales, and under both LoRA and full fine-tuning regimes (Tables 2 and 3). This is a concrete, actionable result and it is demonstrated across a wide enough set of conditions to be credible.

The comparison against regression-based DMO baselines (RegMix-style) is also well-evidenced. The paper shows that the merged proxy needs K training runs (i.e., one per domain expert). This proxy model matches or outperforms regressors fit on 15-16+ full fine-tuning runs, and for Intern3.5-VL, the regressors plateau below the proxy's correlation even at 4× the compute. This efficiency argument is clearly stated and the supporting figures (Fig. 2) make the comparison easy to follow.

The cross-budget experiment (Appendix C.2) further strengthens the claims by showing that experts trained at 50k samples merge into proxies that correlate well (Spearman R = 0.72–0.75) with 100k-budget mixture-trained models, while 10k-trained experts correlate poorly (R = 0.28–0.40). It's important to note that this kind of ablation adds meaningful nuance to the practical deployment advice and speaks to the paper's rigor.

The theoretical justification in Section 5, while relying on simplifying assumptions, is clearly laid out and the assumptions (local convexity, uniform Hessians) are explicitly stated rather than hidden. The paper provides supporting empirical evidence (Figs. 3–4) to validate these assumptions, lending at least plausibility to the theoretical framing.

### Where the evidence needs to be argued better

1. **Correlation framing vs. selection results**: The abstract and introduction lean on the framing of "high rank correlation" (Spearman R = 0.57–0.78), but this is a moderate range rather than an exceptionally strong one. The more convincing evidence is the downstream selection accuracy (~1% from optimum in most cases), and the paper would benefit from leading with that more prominently. As it stands, the correlation claim as written could cast doubt in the minds of readers, especially those familiar with the variance in Spearman R at small sample sizes.

2. **Theoretical gap for K > 2**: The empirical support for the linear mode connectivity and local convexity assumptions (Figs. 3–4) is only demonstrated for the K=2, General+OCR pair. Extending these visualizations to K=3 or K=4 domain combinations would significantly strengthen the claim that the theoretical underpinning generalizes to the full experimental scope of the paper.

3. **Statistical robustness**: The "Selected" mixture results (the paper's primary claim about selection accuracy) are reported with only 3 seeds, and the regression baseline comparison draws from a population of just 24 mixtures. This is also acknowledged by the authors stating that they are not fully conclusive. More seeds, or a sensitivity analysis, would add weight to these comparisons and better support the claims made.

4. **Generalist setting caveat**: The paper is honest that DMO matters less in *generalist* settings (since median and best mixtures are close to begin with), but this somewhat undercuts the scope of the headline claim. This nuance needs to be argued more clearly upfront so readers understand the method is primarily motivated by and strongest in the specialist setting.

### Closing note

The paper is clearly written, the evidence is presented in a well-organized fashion, and the authors are commendably transparent about the limitations of their approach. The gaps above are areas where the framing or additional analysis could better support claims that are, at their core, substantively correct. This is not a paper that makes unsupported bold statements, but one where certain claims need to be contextualized or empirically broadened. To that end, the paper largely meets TMLR's evidence standard, with the caveats noted above needing to be addressed or better argued.

**Requested Changes:**

The paper is in good shape overall and the core contributions are solid. However, there are some changes that could strengthen the work.

### Critical

1. **Extend theoretical validation to K > 2 domain pairs** (Section 5, Figs. 3–4): The empirical support for the local convexity and linear mode connectivity assumptions, which are the theoretical backbone of the method, is currently only demonstrated for the K=2, General+OCR domain pair. Given that the main experiments span K = 2, 3, and 4 domains, the theoretical validation needs to keep pace. Extending Figs. 3 and 4 to at least one K=3 or K=4 configuration, or to a different K=2 domain pair, would significantly strengthen the claim that the theoretical underpinning generalizes across the experimental scope of the paper. As it stands, this gap could potentially cast a doubt that needs to be preemptively addressed.

2. **Reframe the correlation claim in the abstract and introduction**: The abstract leads with "high rank correlation" (Spearman R = 0.57–0.78), but this is a moderate range, and readers familiar with Spearman R at small sample sizes may find this framing hard to accept at face value. It's important to note that the more convincing evidence here is the proxy-selected mixture landing within ~1% of the grid-search optimum in the majority of specialist cases, and this should be the primary claim foregrounded in the abstract. The correlation numbers are better positioned as supporting diagnostics rather than the headline result. This reframing would make the paper's strongest evidence immediately visible to the reader.

3. **Address statistical robustness of the "Selected" results**: The primary mixture-selection results (Tables 2 and 3) are reported over only 3 seeds. Given that these results are the core empirical claim of the paper, reporting variance or confidence intervals across additional seeds, or at minimum discussing the sensitivity of the selected mixture to seed variation, would add weight to these comparisons. The authors themselves acknowledge the regression baseline comparison is not fully conclusive due to the small population (24 mixtures). Elaborating on this discussion explicitly and how it affects the conclusions would be beneficial for readers.

### Non-Critical (would strengthen the work)

4. **Clarify the novelty delta relative to Wang et al. (2026)**: The concurrent/independent work by Wang et al. (2026), which also uses model merging as a DMO proxy but in the text-only LLM setting, is discussed in the related work section. The distinction that this paper addresses the multimodal setting could be stated more explicitly upfront (e.g., in the introduction or abstract), to preempt novelty questions during review and make the paper's positioning clearer to readers encountering both works.

5. **Discuss or explore continuous mixture search**: The paper restricts itself to a finite grid of mixture weights and does not explore gradient-based or iterative refinement of the mixture weights in the proxy space. Even a brief discussion of whether the merged proxy is differentiable with respect to the mixture weights w, and whether gradient-based search is feasible in principle, would broaden the paper's scope and pave the way for future research in this direction.

---

> ### Author Response · Authors · 2026-07-10
>
> We thank the reviewer for their positive and detailed assessment. We address each point below.
>
> **(C1, W2) Extend theoretical validation to other domain pairs**
>
> As suggested, we visualize the loss functions in the neighbourhood of expert models (Fig.3) also for the two missing domains (Chart Understanding, Visual Perception). We also reproduce Fig. 4 for the other 5 possible pairs of four domains. Overall, these new visualizations support the local convexity assumptions and further verify the observations in the main document. We add these figures and related discussions in the paper.
>
> **(C2, W1) Reframe the correlation claim**
>
> We thank the reviewer for this helpful suggestion. We agree that the strongest evidence for the practical usefulness of the method is the quality of the mixture selected by the proxy. Therefore, we have modified the abstract accordingly by replacing *“... merged proxy models exhibit a high rank correlation with models trained on actual data mixtures.”* with *“... merged proxy models consistently select near-optimal mixtures.”*. Similarly, we updated the second contribution listed at the end of the introduction.
>
> **(C3, W4) statistical robustness**
>
> As requested, we repeat the main experiments with two additional seeds and report the updated results over 5 total seeds in the updated document. We do not observe any significant changes in the results.
> Also, to improve the statistical significance of the experiments comparing against regressor-based methods, we add 8 additional mixtures to the current batch of 24, and increase the size of the validation set from n=8 to n=16. We also increase the number of random seeds from 100 to 1000. The updated plots are in the revised document, and the main outcomes remain unchanged.
>
> **(C4, MW) novelty delta relative to Wang et al.**
>
> We thank the reviewer for the constructive suggestion. We follow it and state the distinction from the concurrent work also in the Introduction. We added the following sentence at the end of the third paragraph: *“Previous work explored the idea of parameter merging as a proxy for data mixing in unimodal tasks (Maldonado et al., 2024; Tao et al., 2025; Wang et al., 2026). Instead, we focus on multimodal LLM supervised fine-tuning, where data sources correspond to heterogeneous vision-language domains.”*
>
> Additionally, to better position our work into the literature, we added a new section “Model Merging and Data Mixing” to the Related Work where we discuss existing literature investigating the relation between parameter merging and data mixing.
>
> **(C5, W3) Discuss or explore continuous mixture search**
>
> We agree that exploring continuous optimization over mixture weights is an important direction for improving practical applicability.
>
> Under linear merging, $\theta^M_w = \sum_{i=1}^{K} w_i \theta_i$, $w \in \Delta^{K-1}$,
> the proxy parameters are continuous and differentiable with respect to $w$ on the simplex. Thus, if the optimization objective is differentiable, one could in principle solve the DMO optimization problem in Eq. 4 using techniques such as projected gradient descent over the simplex.
>
> In this work, we use finite candidate sets because our main goal is to compare merged-proxy selection against the corresponding mixture-trained grid-search oracle in a controlled way. Moreover, our primary objectives are downstream benchmark scores, which are mostly based on exact-match style metrics after decoding, and are non-differentiable with respect to $w$.
> As requested, we perform a preliminary experiment evaluating the effectiveness of optimizing over the full simplex of merging weights. Specifically, we fit a regressor on the merged models' accuracies and then optimize over the regressed scores. Results indicate that optimizing over the continuous probability simplex can indeed lead to further improvements. We report these results in Sec. C.7 of the updated supplementary material.
>
> **Generalist setting caveat**
>
> We follow the suggestion and upfront in the revised Introduction that *“[...] we empirically observe that the DMO problem is more relevant on specialist objectives, where selecting the optimal mixture can lead to significant gains in performance. Instead, we find that differences among mixtures are moderate on the generalist objective, since gains on some tasks are offset by losses on others.”*

---

### Review · Reviewer_sw1J · 2026-06-30

**Summary Of Contributions:**

The paper proposes to use model merging for multimodal data mixture selection.
The approach is intended to alleviate the high compute cost that is associated with finding good weights for data mixes which is an ongoing area of research with high impact.
Across various experiments the authors show that model merging can effectively find good mixture coefficients.
Finally, a justification based on local approximations is provided.

Strengths

- The paper is very well-written and easy-to-read, I did not have troubles understanding any of the parts

- The distinction between specialist and generalist models is interesting and it is a valuable finding that model merging works well for both settings

- The experiments are vast and the publicly released checkpoints can be useful for future research

Weaknesses

-  There have already been works that use model merging for data mixes which also discuss the relationship between model merging mixture weights and joint training mixture weights more clearly. I believe it would be useful to discuss these more clearly, as described in my requested changes. Under this lens it would be useful to understand the exact contribution of the analysis in Sec. 5 better.

**Additional Comments:**

- Minor: Should Eq. 1 really be a set union or is it usually intended that example counts are kept?

- Minor: Alg. 1 is a little bit small

- Is linear merging always sufficient for rank correlation? Shouldn't there intuitively be cases where the approximation obtained via linear merging is too coarse?

- The correlation in Fig. 1 seems to depend a lot on the chosen model, could you comment on this more?

- Did you try using the closed form solution of Eq. 10 for merging?

**Audience:**

Yes

**Audience Explanation:**

- Data mixture optimization is an important problem that is relevant to virtually all frontier model training. Many works have been dedicated to the topic, which the authors also discuss, and any progress on it can lead to significant cost savings and positive impact.

- The relationship to model merging is known in prior work, as also discussed in other parts of this review, but the model merging community is still likely to find the work interesting, as it discusses a) a new domain in multimodal models and b) an interesting distinction of comparing specialist vs. generalist models.

**Broader Impact Concerns:**

I do not think there are any concerns about the broader impact of this work.

**Claims And Evidence:**

Yes

**Claims Explanation:**

- The experiments are vast, spanning various model architectures, datasets, and comparisons to other approaches like regressor-based data mix optimization (though this particular comparison could be extended as the findings were not entirely conclusive, as acknowledged by the authors)

- The experiments clearly show that performance of mixes obtained via merging and via finding the optimal data mix via grid search match well

- The experiments show that model merging can be used to optimize for both generalist and specialist data mixes

**Requested Changes:**

- The claim that the work "establish[es] model merging as an effective surrogate for data mixture evaluation" could be worded more carefully in the context of related works that have already proposed model merging for data mixture optimization prior to the submitted manuscript (Maldonado et al. 2024). The work also discusses conditions under which $\theta_w^\ast$ is more closely related to the merged model $\theta_w^M$. I believe a discussion of the work and how it relates to the submitted manuscript would be useful. The work also makes the connection between merging and mixing coefficients more explicit.

- The method also seems quite closely related to Li et al. 2025 who use quadratic surrogates. It would be good to discuss their connection in more detail.

- The closed-form solution in Eq. 10 is reported in multiple works, for example, Maldonado et al. 2024 and Daheim et al. 2024, where it is also used to improve model merging. There is also a relation to Fisher-weighted averaging (Matena et al. 2022) which would be useful to discuss. The linear combination without Hessians is also better seen as using isotropic Gaussians than assuming that all Hessians are the same, as this sheds light on the more simple approximation that is made. Finally, it would be useful if the authors discussed trade-offs of using simpler approximations vs. using Hessians for merging.

- The claim that Hessians are not feasible to compute is not correct, as there exist e.g. natural-gradient-based variational learning algorithms that estimate the Hessian during training (though not the full Hessian), for example, IVON (Shen et al. 2024).

## References

Daheim, Nico, et al. "Model merging by uncertainty-based gradient matching." ICLR (2024).

Li, Lu, et al. "Map: Low-compute model merging with amortized pareto fronts via quadratic approximation." International Conference on Learning Representations. Vol. 2025. 2025.

Maldonado, Hugo Monzón, et al. "How to Weight Multitask Finetuning? Fast Previews via Bayesian Model-Merging." arXiv preprint arXiv:2412.08147 (2024).

Matena, Michael S., and Colin Raffel. "Merging models with fisher-weighted averaging." Advances in Neural Information Processing Systems 35 (2022): 17703-17716.

Shen, Yuesong, et al. "Variational learning is effective for large deep networks." ICML (2024).

---

> ### Author Response · Authors · 2026-07-10
>
> We thank the reviewer for the constructive comments and address them below.
>
> **(C1,C2) Better positioning into the literature**
>
> We agree that the claim *"(our work) establish[es] model merging as an effective surrogate for data mixture evaluation"* should be better phrased, especially w.r.t. prior work exploring model merging for data mixture selection. We softened this claim throughout the paper to more precisely state our core contribution: extending and validating the idea of model merging as a surrogate for DMO in the multimodal SFT setting, at a wide empirical scale not previously explored (across MLLM families, model sizes, numbers of domains, data budgets, optimization objectives), together with a distinction between specialist and generalist objectives, and a comparison against regression-based baselines. Specifically, in the revised document, we (i) rephrase the referred claim to *“providing large-scale evidence that simple linear merging is an effective surrogate for multimodal data-mixture selection”* (ii) clarify in the introduction that previous work explored the idea of parameter merging as a proxy for data mixing (iii) add a new section “Model Merging and Data Mixing” to the Related Work where we discuss existing literature investigating the relation between parameter merging and data mixing.
>
> In the revised Related Work, we also discuss the cited study from Maldonado et al. (2024), which explicitly investigates how domain weights in multitask fine-tuning relate to Bayesian model-merging coefficients. Our work is complementary: rather than proposing a new Bayesian merging rule, we study whether the simplest weight-aligned linear merge is sufficient as a surrogate for multimodal data mixture optimization in MLLM SFT. We evaluate this question across multiple MLLM families, model scales, fine-tuning regimes, domain counts, and both specialist and generalist targets.
>
> The reviewer also cites MAP (Li et al., 2025), a quadratic approximation surrogate to efficiently identify Pareto-optimal model-merging coefficients. While related, MAP has a different objective from ours: it optimizes merging coefficients to construct merged models on a Pareto front, while our problem setting aims at optimizing data mixtures for supervised fine-tuning. We include Li et al. (2025) in the Limitations and Future Work section of the revised document as a possible alternative to grid-search optimization over merging weights.
>
> **(C3,C4) Equation 10 and Section 5 analysis**
>
> We would like to clarify that the purpose of our analysis in Section 5 is not to claim a novel closed-form merging solution, but to provide intuition for why a merged proxy can preserve the ranking of static mixture-trained models. We revise this section to explicitly clarify that similar closed-form expressions appear in previous works (Maldonado et al. , 2024, Daheim et al., 2024, Matena & Raffel, 2022), and we clarify that our analysis is used to connect these ideas to our merged proxy for multimodal DMO, and not to present Eq. 10 as a new result.
> Also, we followed the reviewer’s suggestion and explained that our derivation can be equivalently framed in a Bayesian setting, in which linear merging follows from approximating each expert’s local posterior with an isotropic Gaussian of equal precision.
>
> Finally, we revised the statement that Hessians are “not computationally feasible” to a more precise one: computing and storing full dense Hessians for modern MLLMs is infeasible, but there exist techniques to estimate curvature information. We will add Shen et al. (2024) and discuss that curvature-aware merging could improve the proxy but introduces additional computational and engineering costs. This motivates our choice of simple linear merging: it is easy to implement, architecture-agnostic, and empirically sufficient for effective mixture optimization in our experiments.
>
> **Extend comparison with regressor-based methods**
>
> To improve the statistical significance of the experiments comparing against regressor-based methods, we add 8 additional mixtures to the current batch of 24, and increase the size of the validation set from n=8 to n=16. We also increase the number of random seeds from 100 to 1000. The updated plots are in the revised document, and the main outcomes remain unchanged, e.g., that the merged proxies are approximately $4\times$ more efficient than regressors, as they require only a fraction of training runs to provide identical reliability.

---

> > ### Author Response · Authors · 2026-07-10
> >
> > **Minor comments**
> >
> > Thank you for the suggestions. We address the minor comments as follows:
> >
> > - *Should Eq. 1 really be a set union?*
> > We agree that Eq. 1 is formally incorrect, but we believe it is intuitive and easy to interpret, given the nearby definition. As the reviewer correctly noted, the mixture dataset should preserve example counts rather than collapse repeated samples as in a set union; in the revised document, we remove this ambiguity by specifying that the samples are disjoint.
> > - *Alg. 1 is small.* We enlarge Alg. 1 in the revised version for readability.
> > - *Is linear merging is always sufficient?* No, this is not guaranteed in general, and we don't intend to claim it is. The analysis in Section 5 identifies specific conditions under which linear merging is a good approximation: local convexity of the loss around the experts combined with the uniform/isotropic-curvature simplification. These conditions are not necessary for effective DMO, but they suggest when the merged proxy could fail: (i) when distances between experts are very large, making the second-order Taylor approximation break down, or (ii) when per-expert curvature across domains is very different (violating the uniform-Hessian/isotropic assumption). These are exactly the kinds of settings where curvature-aware or more sophisticated merging strategies may improve the proxy.
> > - *Model-dependence of correlation in Fig. 1.* We believe this connects to Intern3.5-VL's pretraining mixture, already containing substantial instruction-tuning data. This causes different loss geometry around the SFT optimum compared to Qwen2-VL, whose pretraining does not include such data.
> > - *Using the closed-form solution in Eq. 10.* We did not use Eq. 10 since approximating the (full) Hessians is impractical, introducing extra compute and implementation complexity. We intentionally use simple linear merging because the goal is to test whether a simple, low-overhead proxy is already sufficient for ranking mixtures. We discuss this in the first paragraph of the Limitations and Future Work section.

---

> > ### Comment · Reviewer_sw1J · 2026-07-22
> >
> > Thank you for your responses, I would like to address one point:
> >
> > > "While related, MAP has a different objective from ours: it optimizes merging coefficients to construct merged models on a Pareto front, while our problem setting aims at optimizing data mixtures for supervised fine-tuning."
> >
> > Would the goal of finding the best data mixture not be to find the Pareto front of multitask training? Initially the objectives seem to be the same to me, could you please clarify this point?
> >
> > Otherwise this response and your next response clarify my questions.

---

### Review · Reviewer_hSmL · 2026-06-30

**Summary Of Contributions:**

This work proposes model merging as a proxy for ranking different possible mixture weights for mixing data from different datasets to optimize some target tasks. They first train only individual expert models per dataset. Then, for evaluating any mixture weights, they approximate the model by the linear combination of expert models using those weights and evaluate the performance with the resulting model. The main focus of this paper was for multimodal fine-tuning data mixture optimization (DMO).

Strengths:
1. Comprehensive experiments with different data budgets, tasks, and fine-tuning methods.
2. An efficient and practical method for finding the data mixture that is close to the best data mixture achieved by grid search.
3. This work is timely and addresses a useful, understudied problem. While DMO has been explored mostly for text-only LLMs, the multimodal setting has received less attention, and this paper helps fill the gap.

Weaknesses:
1. The biggest weakness of this approach is that it only gives us a final static mixture weight for data mixing, which might be suboptimal. Recent works [1-4] have proposed dynamic mixing methods, where we might have different mixture weights over time during the training (e.g. per iteration or epoch).
2. Limited novelty compared to the Merge To Mix paper.
3. Marginal practical gain in the average performance for both the authors’ method and the best achieved by grid search, which again shows the importance of dynamic DMO.

[1] Dynamic Data Mixing Maximizes Instruction Tuning for Mixture-of-Experts
[2] Efficient Online Data Mixing For Language Model Pre-Training
[3] RegMix-D: Dynamic Data Mixing via Proxy Training Trajectories
[4] TiKMiX: Take Data Influence into Dynamic Mixture for Language Model Pre-training

**Audience:**

Yes

**Audience Explanation:**

Finding the optimal data mixture weights for multimodal fine-tuning is a timely and active research area. Moreover, by publishing their code and checkpoints, they will pave the way for further contributions in the multimodal setting.

**Broader Impact Concerns:**

The impact statement is sufficient.

**Claims And Evidence:**

Yes

**Claims Explanation:**

The authors have provided comprehensive experiments for multimodal DMO in both full fine-tuning and LoRA finetuning scenarios. The results demonstrate the efficiency and efficacy of model merging as a proxy for ranking different possible data mixtures in terms of the final performance. They have also included a theoretical justification for their method as well.

**Requested Changes:**

In the limitation section, it would be better to also state that your current framework is only relevant for static data mixture optimization and might be suboptimal compared to methods that dynamically change the mixture weights over time during training.

For equation 7 in section 5.1, it would be better to derive why the loss on the data mixture, which is sum of the losses on each of the N data samples sampled from {D_i}_i=1^K datasets with probability distribution (w_1,...,w_i,...,w_K), is equivalent to weighted sum of losses on the whole D_i datasets where weights are w_i values (In expectation).

The following related papers could be cited:
1. Mix Data or Merge Models? Balancing the Helpfulness, Honesty, and Harmlessness of Large Language Model via Model Merging
2. Mix Data or Merge Models? Optimizing for Diverse Multi-Task Learning
3. Multi-task Code LLMs: Data Mix or Model Merge?
4. Towards a Data-Parameter Correspondence for LLMs: A Preliminary Discussion
5. An Empirical Study of Multimodal Model Merging
6. MaD-Mix: Multi-Modal Data Mixtures via Latent Space Coupling for Vision-Language Model Training

---

> ### Author Response · Authors · 2026-07-10
>
> We thank the reviewer for their positive assessment and address all concerns below.
>
> **(W1) Only static optimization**
>
> We agree that allowing the mixture weights to vary throughout training could lead to a better final model. In the revised document, we added a third paragraph to the Limitations and Future Work section where we explicitly state our focus on static mixing.
>
> Nevertheless, static mixture optimization as considered in our work is still largely useful in practice.
> Dynamic mixing commonly targets language model pretraining (as in the cited works [2], [3], [4]), where a model sees a very large number of steps/tokens, and the optimal mixture weights can shift significantly over the course of training. Our work instead targets multimodal SFT, which typically involves fewer steps and a relatively smaller instruction-tuning corpus. In this regime, state-of-the-art MLLMs (Tong et al., 2024; Deitke et al., 2025) still rely on a fixed mixture ratio decided before training. The practical utility of dynamic reweighting over such short training horizons is less established and requires additional engineering complexity. Therefore, static mixture selection is still a practically relevant problem setting for SFT, and improving it, as we do, has immediate applicability to how MLLMs are trained today.
>
> Furthermore, we note that the core mechanism we propose can be naturally extended to dynamic mixture selection by decomposing training into stages and using the merged proxies to choose the mixture for each stage. Alternatively, one could also use the selected static mixture as a strong initialization for dynamic methods. We see these as interesting research directions building directly on our findings.
>
> **(W2) Limited novelty compared to the Merge To Mix paper**
>
> While Merge to Mix (Tao et al., 2025) explores the idea of model merging for DMO, there are several differences:
> Merge to Mix considers a binary problem, i.e., whether a data source should be included in the training mixture. We address the full continuous DMO problem: given $K$ domains, find the mixture weights that maximize performance. This is a harder problem, as it requires optimizing across a larger space of combinations.
> Merge to Mix does not operate at a fixed data budget, as their $w$ could be any vector with entries in \{$0, 1$\}. This confounds data mixing with the benefits of "adding more data" and it is not realistic in MLLM applications.
> We conduct a systematic evaluation of the merged proxy across 14 benchmarks and along several experimental axes. We also compare against regression-based baselines, demonstrating that merged proxies are $4\times$ more efficient. Merge to Mix shows results for CLIP and a single LLM model.
>
> In the Related Work section of the revised document, we extend the comparison with Merge to Mix to better highlight the differences from our work.
>
> **(W3) Marginal Gain**
>
> While the absolute gains in average performance over the uniform/median baselines are modest, we would like to clarify that this is often due to the uniform mixture being already near-optimal. In the paper, we explicitly note this phenomenon (Sec. 4.1, Obs. #2) and hypothesize it is a property of averaging over a broad, heterogeneous set of benchmarks: once a mixture has "sufficient task coverage" across all target domains, small changes in mixture weights have limited effect on the average, since gains on some benchmarks are offset by losses on others. This is a property of the generalist objective itself, not necessarily evidence that the underlying optimization landscape varies significantly throughout training. Importantly, this behavior is much less pronounced in the specialist setting, where mixture choice matters substantially more, and merged proxies often select mixtures close to the exact-search optimum. For example, in the 4-domain setting, the selected mixture is within 1 percentage point of exact search in 11/14 cases for Qwen2-VL-2B and 10/14 cases for Intern3.5-VL-2B.
>
> In the revised Introduction, we describe this empirical finding about the difference in available gains between the generalist and specialist objectives.
>
> **Other changes**
>
> We thank the reviewer for the suggestions. We agree with all three suggestions and incorporated them as follows:
> - As discussed above, we explicitly state in the Limitations and Future Work section that our focus is on static data mixture optimization, and that extensions to dynamic selection could yield better results.
> - We clarify that Eq. 7 should formally be read in expectation over the mixture-sampling process, and we will make its derivation explicit.
> - Thank you for these pointers. Several are directly relevant, and we incorporate them into the Related Work.
>
> ---
>
> References:
>
> Tong, et al. “Cambrian-1: A fully open, vision-centric exploration of multimodal llms.” NeurIPS, 2024.
>
> Deitke, et al. “Molmo and pixmo: Open weights and open data for state-of-the-art vision-language models.” CVPR, 2025.

---

> > ### Comment · Reviewer_hSmL · 2026-07-22
> >
> > Thank you authors for addressing my concerns. I have no further comments.